# An Information-Theoretic Analysis of Thompson Sampling for Logistic Bandit Problems

**Amaury Gouverneur**                                                        *amauryg@kth.se*
*Division of Information Science and Engineering (ISE)*
*KTH Royal Institute of Technology*
*Stockholm, Sweden*

**Tobias J. Oechtering**                                                       *oech@kth.se*
*Division of Information Science and Engineering (ISE)*
*KTH Royal Institute of Technology*
*Stockholm, Sweden*

**Mikael Skoglund**                                                          *skoglund@kth.se*
*Division of Information Science and Engineering (ISE)*
*KTH Royal Institute of Technology*
*Stockholm, Sweden*

**Reviewed on OpenReview:** *https: // openreview. net/ forum? id=94y5XfiJ7N*

## Abstract

We study the performance of the Thompson Sampling algorithm for logistic bandit problems. In this setting, an agent receives binary rewards with probabilities determined by a logistic function, $\exp(\beta\langle a, \theta\rangle)/(1 + \exp(\beta\langle a, \theta\rangle))$, with parameter $\beta > 0$, and both the action $a \in \mathcal{A}$ and the unknown parameter $\theta \in \mathcal{O}$ lie within the $d$-dimensional unit ball. Adopting the information-theoretic framework introduced by Russo & Van Roy (2016), we derive regret bounds via the analysis of the information ratio, a statistic that quantifies the trade-off between the immediate regret incurred by the agent and the information it just gained about the parameter $\theta$. We improve upon previous results and establish that the information ratio is bounded by $d(4/\alpha)^2$, where $d$ is the dimension of the problem and $\alpha$ is a *minimax measure* of the alignment between the action space $\mathcal{A}$ and the parameter space $\mathcal{O}$. Notably, our bound does not scale exponentially with the logistic slope and is independent of the cardinality of the action and parameter spaces. Using this result, we derive a bound on the Thompson Sampling expected regret of order $O(d\alpha^{-1}\sqrt{T\log(\beta T/d)})$, where $T$ is the number of time steps. To our knowledge, this is the *first regret bound for any logistic bandit algorithm* that avoids any exponential scaling with $\beta$ and is independent of the number of actions. In particular, when the parameters are on the sphere and the action space contains the parameter space, the expected regret bound is of order $O(d\sqrt{T\log(\beta T/d)})$.

## 1 Introduction

This paper studies the logistic bandit problem, where an agent sequentially interacts with an unknown environment with parameter $\theta \in \mathcal{O}$. At each time step, the agent selects an action $a \in \mathcal{A}$ and receives a binary reward whose probability of being one is given by the logistic function $\exp(\beta\langle a, \theta\rangle)/(1 + \exp(\beta\langle a, \theta\rangle))$ with slope parameter $\beta > 0$. In this setting, both the action space and the parameter space are closed bounded subsets of the $d$-dimensional real space $\mathbb{R}^d$. The goal of the agent is to maximize its total reward, or equivalently to *minimize its regret*, that is, the difference between the optimal cumulative reward and the cumulative reward achieved by the agent. This setting is used to model various scenarios, such as click-through rate prediction, spam email detection, and personalized advertisement (Chapelle & Li, 2011; Russo

& Van Roy, 2018). Beyond these examples, bandit-based decision systems are also used in news and content recommendation, dynamic pricing, information retrieval, and healthcare (see Bouneffouf et al. (2020) for a survey of the different applications, and Li et al. (2010) for a large-scale deployment).

The performance, or regret, of algorithms for logistic bandits has been extensively studied, with significant contributions including analyses of Upper Confidence Bound (UCB) algorithms (Filippi et al., 2010; Li et al., 2017; Faury et al., 2020) as well as the study of Thompson Sampling (TS) (Russo & Van Roy, 2014; Dong et al., 2019; Abeille & Lazaric, 2017). However, all existing regret bounds for logistic bandits exhibit an exponential dependence on the parameter $\beta$ (see Table 1) or scale poorly with the number of actions. Those results are unsatisfactory because, in practice, the distinction between near-optimal and sub-optimal actions gets more pronounced as $\beta$ increases, which can make it easier to find near-optimal actions. Similarly, algorithms like TS have demonstrated strong empirical performance even in problems with large or continuous action spaces (Russo & Van Roy, 2014). This gap between theoretical bounds and empirical behavior was already pointed out by McMahan & Streeter (2012), who called for improved regret analyses in the logistic bandit setting, a challenge that has long remained open.

In this work, we focus on the Thompson Sampling algorithm (Thompson, 1933), which, despite its simplicity, has proven to be effective for a wide range of problems (Russo et al., 2018; Chapelle & Li, 2011). Analyzing the TS expected regret, Russo & Van Roy (2016) introduced the concept of the information ratio, defined as the ratio between the squared expected instantaneous regret and the information gained about the optimal action from the current observation. Building on this framework, Dong & Van Roy (2018) derived a near-optimal regret rate of $O(d\sqrt{T \log T})$ for $d$-dimensional linear bandit problems. However, applying this analysis to the logistic setting yields regret bounds that grow exponentially with the parameter $\beta$ (see Appendix E). Using numerical simulations, they conjectured that the TS information ratio for logistic bandits scales linearly with the problem's dimension $d$ and is independent of the slope parameter $\beta$ and the cardinality of the action and parameter spaces (Dong & Van Roy, 2018, Conjecture 1).

Table 1: Comparison of various regret guarantees for the logistic bandit problem.

| Algorithm | Regret Upper Bound | Note |
|:---:|:---:|:---:|
| Thompson Sampling (Russo & Van Roy, 2014) | $O\left(e^{\beta} \cdot d \cdot T^{1/2} \cdot \log(T)^{3/2}\right)$ | Bayesian bound |
| GLM-TSL (Abeille & Lazaric, 2017) | $O\left(e^{\beta} \cdot d^{3/2} \cdot \log(d)^{1/2} \cdot T^{1/2} \log(T)^{3/2}\right)$ | Frequentist bound |
| GLM-TSL (Kveton et al., 2020) | $O\left(e^{\beta} \cdot d \cdot T^{1/2} \cdot \log(T|\mathcal{A}|d) + e^{\beta} d^2 \log(T)^2\right)$ | Frequentist bound |
| Logistic-UCB-2 (Faury et al., 2020) | $O\left(d \cdot T^{1/2} \cdot \log(T) + e^{\beta} \cdot d^2 \cdot \log(T)^2\right)$ | Frequentist bound |
| Thompson Sampling (Neu et al., 2022) | $O(d^{1/2} T^{1/2} |\mathcal{A}| \log(\beta T)^{1/2})$ | Bayesian bound |
| Thompson Sampling (*this paper*) | $O\left(d \cdot \alpha^{-1} \cdot T^{1/2} \cdot \log(\beta T/d)^{1/2}\right)$ | Bayesian bound, $\alpha$ is independent of $\beta$ |

Studying specifically the TS regret for logistic bandits, Dong et al. (2019) introduced two statistics to characterize the sets $\mathcal{A}$ and $\mathcal{O}$, the *minimax alignment constant*[1] $\alpha = \min_{\theta \in \mathcal{O}} \max_{a \in \mathcal{A}} \langle a, \theta \rangle$ and the *fragility dimension*[2] $\eta$, which is the cardinality of the largest subset of parameters such that their corresponding optimal action is "misaligned" (i.e. the inner product is negative) with any other parameter from the subset. Using those statistics, they state the TS information ratio is bounded by $100 \max(d, \eta)\alpha^{-2}$. This result is unsatisfying as in general, the fragility dimension can grow exponentially with the dimension $d$, but more problematic, their regret analysis is incorrect as it relies on the rate-distortion bound from Dong & Van Roy (2018), which is incompatible with a bound on the TS information ratio. Additionally, their analysis of the information ratio bound is only rigorous for $\beta \leqslant 2$ as it relies on an unproven inequality for larger values of $\beta$. We elaborate in more detail on these gaps in Appendix F.

---

[1] This statistic is referred to as the *worst-case optimal log-odds* in the work of Dong et al. (2019).
[2] The definition can be found in Dong et al. (2019, Definition 2).

In this paper, we address the above limitations and obtain a regret bound for logistic bandits that scales linearly in the problem dimension $d$ while not scaling exponentially with $\beta$ or depending on the cardinality of the action and parameter spaces. Importantly, our bound does not require introducing the fragility dimension. Our main contributions are as follows:

- We prove an information-theoretic regret bound of order $O(\sqrt{T\Gamma(\mathrm{H}(\Theta_\varepsilon) + \beta^2\varepsilon^2 T)})$ that holds for infinite and continuous action and parameter spaces. The bound relies on $\mathrm{H}(\Theta_\varepsilon)$, the entropy of the parameter quantized at scale $\varepsilon$, and on the average expected TS information ratio, $\Gamma$.

- We present a new analysis showing that, for all $\beta > 0$, the TS information ratio for logistic bandits is bounded by $d(4/\alpha)^2$ where $\alpha$ is a *minimax alignment constant* between $\mathcal{A}$ and $\mathcal{O}$ which is defined as $\alpha = \min_{\theta \in \mathcal{O}} \max_{a \in \mathcal{A}} \langle a, \theta \rangle$. Notably, our bound does not depend on the fragility dimension.

- We prove a regret bound of order $O(d/\alpha\sqrt{T\log(\beta T/d)})$ for Thompson Sampling. To our knowledge, this is the *first regret bound for any logistic bandit algorithm* that does not scale exponentially with $\beta$ and is independent of the number of actions $|\mathcal{A}|$.

- Additionally, we show that if the parameters are on the sphere and the action space encompasses the parameter space, the expected regret of Thompson Sampling is bounded in $O(d\sqrt{T\log(\beta T/d)})$.

These results help to explain the empirical performance of Thompson Sampling in logistic bandit problems across different logistic regimes and for large or continuous action spaces, and they highlight the importance of alignment between the action and parameter spaces.

The rest of the paper is organized as follows. Section 2 introduces the logistic bandit problem, defines the Bayesian expected regret, and the specific notation used. In Section 3, we introduce the Thompson Sampling algorithm and the information ratio analysis. Section 4 states and discusses our main results, providing improved regret bounds. Section 5 presents the key ideas for analyzing the information ratio. Then Section 6 illustrates the improvement of our bounds compared to previous regret guarantees through numerical experiments; and finally, Section 7 discusses our results and future extensions.

Throughout the paper, we write random variables with capital letters (e.g., $X$), their realization with lowercase letters (e.g. $x$), and their outcome space in calligraphic letters (e.g., $\mathcal{X}$).

## 2 Problem Setup

In the logistic bandit, an agent interacts sequentially with an environment that is characterized by an unknown parameter $\theta \in \mathcal{O} \subseteq \mathbb{R}^d$. At each time step $t \in \{1, \dots, T\}$, the agent selects an action $a \in \mathcal{A} \subseteq \mathbb{R}^d$ and receives a random reward $R_t \in \{0, 1\}$ sampled from a Bernoulli distribution with probability given by a logistic function applied to the inner product $\langle a, \theta \rangle$,

$$R_t \sim \mathrm{Bern}\Big(\frac{\exp(\beta\langle a, \theta \rangle)}{1 + \exp(\beta\langle a, \theta \rangle)}\Big),$$

where $\beta > 0$ is a scale parameter known to the agent. For the rest of the paper, we denote the logistic function as

$$\phi_\beta(x) := \frac{\exp(\beta x)}{1 + \exp(\beta x)}.$$

In this setting, the action space $\mathcal{A}$ and the parameter space $\mathcal{O}$ are compact subsets of $\mathbb{R}^d$ and, without loss of generality[3], we assume that $\mathcal{A}$ and $\mathcal{O}$ lie within the $d$-dimensional Euclidean unit ball, $\mathbf{B}_d(0, 1)$. For a given action space $\mathcal{A}$ and parameter space $\mathcal{O}$, we define their *minimax alignment constant* as $\alpha := \min_{\theta \in \mathcal{O}} \max_{a \in \mathcal{A}} \langle a, \theta \rangle$. Intuitively, it quantifies how well the action set covers for any of the possible parameters: a value of $\alpha$ close to 1 indicates that, for any parameter, there exists an action that aligns

---

[3]This setting is equivalent to the one considered by Faury et al. (2020) using $\beta > 0$ as the maximal norm for $\theta \in \mathcal{O}$.

well with it, whereas a value close to 0 means that there exists a parameter at best nearly orthogonal to all actions. In the rest of the paper, we assume that the action and parameter spaces are such that $\alpha \geqslant 0$. This assumption is mild. It is already satisfied if the set $\mathcal{A}$ contains two opposite actions $a$ and $a'$ (i.e., $a = -a'$), which ensures $\alpha \geqslant 0$ for any parameter set $\mathcal{O}$.

Following the Bayesian framework, we assume the parameter vector $\Theta \in \mathcal{O}$ is sampled from a known prior distribution $\mathbb{P}_\Theta$. At time step $t \in \{1, \ldots, T\}$, the agent selects an action $A_t$ based on the past observations according to some possibly stochastic decision policy. As the reward distribution depends only on the selected action and the parameter, the random reward $R_t$ is also written as $R(A_t, \Theta)$, where $R : \mathcal{A} \times \mathcal{O} \to \{0, 1\}$ is a stochastic process defined such that $R(a, \theta) \sim \mathrm{Bern}\big(\phi_\beta(\langle a, \theta \rangle)\big)$.

The goal of the agent is to sequentially select actions that maximize the total expected reward, or equivalently, that *minimize the total expected regret* defined as:

$$\mathbb{E}[\mathrm{Regret}(T)] := \mathbb{E}\left[ \sum_{t=1}^{T} R(A^\star, \Theta) - R(A_t, \Theta) \right],$$

where $A^\star$ is the *optimal action* corresponding to the parameter $\Theta$, that is, we set $A^\star = \arg\max_{a \in \mathcal{A}} \mathbb{E}[R(a, \Theta)]$. We define the mapping $\pi_\star(\theta) := \arg\max_{a \in \mathcal{A}} \mathbb{E}[R(a, \theta)]$ so that we can write $A^\star = \pi_\star(\Theta)^4$.

## 3 Thompson Sampling and the Information Ratio

Thompson Sampling is an elegant algorithm for solving bandit problems. It works by randomly selecting actions according to the posterior probability of being optimal. More specifically, at each time step $t \in \{1, \ldots, T\}$, the agent samples a parameter estimate $\hat{\Theta}_t$ from the posterior distribution of $\Theta$, conditioned on the history $H^t$ and selects the action that is optimal for the sampled parameter estimate, $A_t = \pi_\star(\hat{\Theta}_t)$. The pseudocode for the algorithm is given in Algorithm 1. Since the $\sigma$-algebras generated by the history are often used in conditioning, we introduce the notation $\mathbb{E}_t[\cdot] := \mathbb{E}[\cdot | H^t]$ to denote the conditional expectation given the history $H^t$.

---
**Algorithm 1** Thompson Sampling algorithm
---
1: **Input:** parameter prior $\mathbb{P}_\Theta$, mapping $\pi_\star$.
2: **for** $t = 1$ **to** T **do**
3:     Sample a parameter estimate $\hat{\Theta}_t \sim \mathbb{P}_{\Theta | H^t}$.
4:     Take the corresponding optimal action $A_t = \pi_\star(\hat{\Theta}_t)$.
5:     Collect the reward $R_t = R(A_t, \Theta)$.
6:     Update the history $H^{t+1} = H^t \cup \{\hat{\Theta}_t, R_t\}$.
7: **end for**
---

The *information ratio*, originally introduced by Russo & Van Roy (2016), quantifies the trade-off between exploration and exploitation. In this paper, we define the information ratio at time $t$ as the ratio between the *squared instantaneous expected regret* and the *information gained about the environment parameter*,

$$\Gamma_t := \frac{\mathbb{E}_t[R(A^\star, \Theta) - R(A_t, \Theta)]^2}{\mathrm{I}_t(\Theta; R(A_t, \Theta), \hat{\Theta}_t)},$$

where $\mathrm{I}_t(\Theta; R(A_t, \Theta), \hat{\Theta}_t) := \mathbb{E}_t[\mathrm{D}_{\mathrm{KL}}(\mathbb{P}_{R_t | H^t, \hat{\Theta}_t, \Theta} \| \mathbb{P}_{R_t | H^t, \hat{\Theta}_t})]$ is the mutual information between the parameter $\Theta$ and the observed pair $R_t, \hat{\Theta}_t$, *given the history*[5] $H^t$.

This ratio measures the *trade-off* between minimizing the current squared regret and gathering information about the parameter $\Theta$; a small ratio indicates that a substantial gain of information compensates for any significant regret.

---

[4]If multiple actions are optimal for a given parameter, we arbitrarily fix the mapping for that parameter to one of the optimal actions. The mapping $\pi_\star$ is therefore a well-defined function.

[5]This quantity is sometimes referred to as the *disintegrated mutual information*, see for example Negrea et al. (2019).

## 4 Main Results

This section presents our main results on the Thompson Sampling regret for logistic bandits. In Theorem 2, we prove an information-theoretic regret bound for logistic bandits that holds for continuous and infinite parameter spaces. Then, in Proposition 4, we present our key result, a bound on the TS information ratio that depends only on the problem's dimension $d$ and on the minimax alignment constant $\alpha$. By combining this result with our regret bound, we derive our main contribution in Theorem 5, a bound on the expected regret of TS for logistic bandits, which scales as $O(d/\alpha\sqrt{T\log(\beta T/d)})$.

Our first theorem provides a regret bound that holds for large and continuous action spaces. It relies on the entropy of the quantized parameter $\Theta_\varepsilon$, which is the closest approximation to $\Theta$ on an $\varepsilon$-net in the Euclidean space $(\mathcal{O}, \|\cdot\|_2)$. It builds on Gouverneur et al. (2023, Theorem 2) and Neu et al. (2022, Theorem 2) and improves their results for the logistic bandit setting. The proof of Theorem 2 relies on approximating the conditional mutual information $I(\Theta; R_t | \hat{\Theta}_t, H^t)$ with $I(\Theta_\varepsilon; R_t | \hat{\Theta}_t, H^t)$ and bounding the remainder using a Taylor expansion. Importantly, and in contrast to Dong & Van Roy (2018, Theorem 1), this result is compatible with bounds on the information ratio of Thompson Sampling, rather than the *"one-step compressed Thompson Sampling"*. This distinction is crucial as it resolves the incompatibility that arises in the regret analysis of Dong et al. (2019) (c.f. Appendix F).

**Definition 1** *Let the set $\mathcal{O}_\varepsilon$ be an $\varepsilon$-net for $(\mathcal{O}, \|\cdot\|_2)$ with projection mapping $q : \mathcal{O} \to \mathcal{O}_\varepsilon$ such that for all $\theta \in \mathcal{O}$ we have $\|\theta - q(\theta)\|_2 \leqslant \varepsilon$. We define the* quantized parameter *as $\Theta_\varepsilon := q(\Theta)$.*

**Theorem 2** *For all $\beta > 0$, under the logistic bandit setting with logistic function $\phi_\beta(x)$, let the quantized parameter $\Theta_\varepsilon$ be defined as in Definition 1 for any $\varepsilon \in [0,1]$. If the average expected TS information ratio is bounded, $\frac{1}{T}\sum_{t=1}^T \mathbb{E}[\Gamma_t] \leqslant \Gamma$, for some $\Gamma > 0$, then the TS regret is bounded as*

$$\mathbb{E}[\text{Regret}(T)] \leqslant \sqrt{\Gamma T \left( H(\Theta_\varepsilon) + \tfrac{1}{2}\varepsilon^2\beta^2 T \right)}.$$

**Proof 3** *We start by rewriting the TS expected regret using the information ratio:*

$$\mathbb{E}[\text{Regret}(T)] = \sum_{t=1}^T \mathbb{E}[R(A^\star, \Theta) - R(A_t, \Theta)] = \sum_{t=1}^T \mathbb{E}\left[ \sqrt{\Gamma_t I_t(\Theta; R(A_t, \Theta), \hat{\Theta}_t)} \right].$$

*Applying Cauchy-Schwarz for expectations, $\mathbb{E}[\sqrt{X}\sqrt{Y}] \leqslant \sqrt{\mathbb{E}[X]\,\mathbb{E}[Y]}$, followed by Cauchy-Schwarz over the sum, we obtain*

$$\mathbb{E}[\text{Regret}(T)] \leqslant \sum_{t=1}^T \sqrt{\mathbb{E}[\Gamma_t] I(\Theta; R(A_t, \Theta), \hat{\Theta}_t | H^t)} \leqslant \sqrt{\Gamma T \sum_{t=1}^T I(\Theta; R(A_t, \Theta), \hat{\Theta}_t | H^t)}, \tag{1}$$

*where in the last step, we used that by assumption $\sum_{t=1}^T \mathbb{E}[\Gamma_t] \leqslant \Gamma T$. Then, by the chain rule for mutual information (Polyanskiy & Wu, 2025, Theorem 3.7.b) we can write*

$$I(\Theta; R(A_t, \Theta), \hat{\Theta}_t | H^t) = I(\Theta; \hat{\Theta}_t | H^t) + I(\Theta; R(A_t, \Theta) | H^t, \hat{\Theta}_t) = I(\Theta; R(A_t, \Theta) | H^t, \hat{\Theta}_t),$$

*using that $\Theta$ and $\hat{\Theta}_t$ are independent given $H^t$, and that $I(\Theta; \hat{\Theta}_t | H^t) = 0$.*

*Let $\mathbb{P}_{R_t | H^t, \hat{\Theta}_t, \Theta}$ and $\mathbb{P}_{R_t | H^t, \hat{\Theta}_t}$ denote the distribution of $R_t$ conditioned respectively on $H^t, \hat{\Theta}_t, \Theta$ and $H^t, \hat{\Theta}_t$. Then, we have that*

$$I(\Theta; R_t | H^t, \hat{\Theta}_t) = \mathbb{E}_{H^t, \Theta, \hat{\Theta}_t}\left[ \mathbb{E}_{R_t \sim \mathbb{P}_{R_t | \hat{\Theta}_t, \Theta}}\left[ \log \frac{\mathbb{P}_{R_t | \hat{\Theta}_t, \Theta}(R_t)}{\mathbb{P}_{R_t | H^t, \hat{\Theta}_t}(R_t)} \right] \right], \tag{2}$$

*where we used that $\mathbb{P}_{R_t | H^t, \hat{\Theta}_t, \Theta} = \mathbb{P}_{R_t | \hat{\Theta}_t, \Theta}$ since $R_t$ is independent of $H_t$ conditioned on $\Theta$ and $\hat{\Theta}$.*

*Let $\mathcal{O}_\varepsilon$ be an $\varepsilon$-net for $(\mathcal{O}, \|\cdot\|_2)$ with mapping $q : \mathcal{O} \to \mathcal{O}_\varepsilon$, and define $\Theta_\varepsilon := q(\Theta)$ the quantized parameter. We introduce $\overline{\mathbb{P}}_{R_t|H^t,\hat{\Theta}_t,\Theta_\varepsilon}$ to denote the posterior reward distribution given $H^t, \hat{\Theta}_t, \Theta_\varepsilon$, which is obtained by averaging over all $\Theta'$ such that $q(\Theta') = \Theta_\varepsilon$, that is $\overline{\mathbb{P}}_{R_t|\hat{\Theta}_t,\Theta_\varepsilon}(\cdot) := \mathbb{E}\left[\mathbb{P}_{R_t|\hat{\Theta}_t,\Theta'}(\cdot)\Big|q(\Theta')=\Theta_\varepsilon,H^t\right]$.*

*Then, starting from the definition of conditional mutual information, we add and subtract the log-probability $\log \overline{\mathbb{P}}_{R_t|H^t,\hat{\Theta}_t,\Theta_\varepsilon}(R_t)$ inside the inner expectation to obtain*

$$(2) = \mathbb{E}_{H^t,\Theta,\hat{\Theta}_t}\left[\mathbb{E}_{R_t \sim \mathbb{P}_{R_t|\hat{\Theta}_t,\Theta}}\left[\log \frac{\overline{\mathbb{P}}_{R_t|H^t,\hat{\Theta}_t,\Theta_\varepsilon}(R_t)}{\mathbb{P}_{R_t|H^t,\hat{\Theta}_t}(R_t)}\right] + \mathbb{E}_{R_t \sim \mathbb{P}_{R_t|\hat{\Theta}_t,\Theta}}\left[\log \frac{\mathbb{P}_{R_t|\hat{\Theta}_t,\Theta}(R_t)}{\overline{\mathbb{P}}_{R_t|H^t,\hat{\Theta}_t,\Theta_\varepsilon}(R_t)}\right]\right].$$

*Using the law of total expectation, the outer expectation over $\Theta$ may be replaced by one over $\Theta_\varepsilon$, and the first term is exactly the conditional mutual information $\mathrm{I}(\Theta_\varepsilon; R_t \mid H^t, \hat{\Theta}_t)$. We recognize the second term as the expected KL divergence between the true reward distribution $\mathbb{P}_{R_t|\hat{\Theta}_t,\Theta}$ and the quantized posterior $\overline{\mathbb{P}}_{R_t|H^t,\hat{\Theta}_t,\Theta_\varepsilon}$, which by convexity of the KL divergence is at most the average over the quantization:*

$$\mathbb{E}_{H^t,\Theta,\hat{\Theta}_t}\left[D_{\mathrm{KL}}(\mathbb{P}_{R_t|\hat{\Theta}_t,\Theta} \| \overline{\mathbb{P}}_{R_t|H^t,\hat{\Theta}_t,\Theta_\varepsilon})\right] \leqslant \mathbb{E}_{\Theta,\hat{\Theta}_t}\left[\mathbb{E}\left[D_{\mathrm{KL}}(\mathbb{P}_{R_t|\hat{\Theta}_t,\Theta} \| \mathbb{P}_{R_t|\hat{\Theta}_t,\Theta'})\Big|q(\Theta')=\Theta_\varepsilon,H^t\right]\right].$$

*Conditioned on $\Theta = \theta$, $\Theta' = \theta'$, and $\hat{\Theta}_t = \hat{\theta}_t$, we let $\eta = \langle \pi_\star(\hat{\theta}_t), \theta \rangle$, we can write $\eta' = \langle \pi_\star(\hat{\theta}_t), \theta' \rangle$. We then have that $\eta' = \eta + x$ for some $x \in [-2\varepsilon, 2\varepsilon]$. Indeed we have*

$$|\langle \pi_\star(\hat{\theta}_t), \theta \rangle - \langle \pi_\star(\hat{\theta}_t), \theta' \rangle| \leqslant \|\pi_\star(\hat{\theta}_t)\| \cdot \|\theta - \theta'\| \leqslant 1 \cdot (\|\theta - q(\theta)\| + \cdot \|q(\theta) - \theta'\|) \leqslant 2\varepsilon,$$

*where we used the fact that $\mathcal{A} \subseteq B_d(0,1)$, the triangle inequality and the definition of $q(\cdot)$. We can then write the inner KL divergence as*

$$D(x) = \phi_\beta(\eta) \log\left(\frac{\phi_\beta(\eta)}{\phi_\beta(\eta + x)}\right) + (1 - \phi_\beta(\eta)) \log\left(\frac{1 - \phi_\beta(\eta)}{1 - \phi_\beta(\eta + x)}\right).$$

*Differentiating with respect to $x$, and using that $\phi'(x) = \beta\phi_\beta(x)(1 - \phi_\beta(x))$, we have that the first derivative is $D'(x) = \beta(\phi_\beta(\eta + x) - \phi_\beta(\eta))$ and the second derivative is $D''(x) = \beta^2 \phi_\beta(\eta + x)(1 - \phi_\beta(\eta + x)) \leqslant \frac{\beta^2}{4}$. Since $D(0) = 0$ and $D'(0) = 0$, we can use Taylor's remainder theorem and get that*

$$D_{\mathrm{KL}}(\mathbb{P}_{R_t|\hat{\Theta}_t=\hat{\theta}_t,\Theta=\theta} \| \mathbb{P}_{R_t|\hat{\Theta}_t=\hat{\theta}_t,\Theta'=\theta'}) = D(x) = \int_0^x (x-t)D''(t)\mathrm{d}t \leqslant \frac{1}{2}\frac{\beta^2}{4}x^2 \leqslant \frac{1}{2}\beta^2\varepsilon^2.$$

*Summing over $t$ and applying the chain rule for conditional mutual information, we obtain*

$$\sum_{t=1}^T \mathrm{I}(\Theta_\varepsilon; R_t \mid H^t, \hat{\Theta}_t) = \mathrm{I}(\Theta_\varepsilon; H^T) \leqslant \mathrm{H}(\Theta_\varepsilon).$$

*Collecting the $\frac{1}{2}\beta^2\varepsilon^2$ terms and inserting the above bound into eq. (1) yields*

$$\mathbb{E}[\mathrm{Regret}(T)] \leqslant \sqrt{\Gamma T \left(\mathrm{H}(\Theta_\varepsilon) + \tfrac{1}{2}\beta^2\varepsilon^2 T\right)},$$

*which is the desired result.*

In the following, we present an important proposition; we prove an upper bound on the TS information ratio that depends only on the problem dimension $d$ and the minimax alignment constant $\alpha$. Notably, this upper bound is independent of the logistic slope $\beta > 0$ as well as of the cardinality of the action and parameter spaces, and depends linearly on the the problem's dimension $d$. Those properties were anticipated in Dong & Van Roy (2018, Conjecture 1).

**Proposition 4** *For all $\beta > 0$, and for all $\mathcal{A}, \mathcal{O} \subseteq \mathbf{B}_d(0,1)$ with minimax alignment constant $\alpha$, under the logistic bandit setting, with logistic function $\phi_\beta(x)$, the TS information ratio is bounded as*

$$\Gamma_t \leqslant \frac{16d}{\alpha^2}.$$

The dependence on $1/\alpha$ should not be surprising: when the action and parameter spaces are poorly aligned, that is when $\alpha$ is small, it is possible to construct an action space $\mathcal{A}$ and a parameter space $\mathcal{O}$ such that the parameter vectors are nearly orthogonal to all actions (see Dong et al. (2019, Appendix D)). In such a case, the reward probabilities associated with different actions become almost indistinguishable, so observing the reward provides little information about the environment, which in turn leads to larger information ratios.

The proof techniques used for Proposition 4 build upon and innovate over prior work in two key ways. Our first innovation is to control the information ratio by relating both regret and information gain to the *expected variance of the regret*, conditioned on the sampled parameter; lower bounding the information gain using the regret variance instead of the regret expectation as in Russo & Van Roy (2016) or Dong & Van Roy (2018), which makes it possible to avoid scaling with the smallest slope of the logistic function which decreases exponentially with the parameter $\beta$. Our second innovation is to show that the limit case $\beta \to \infty$ can *serve as a uniform upper bound*, thereby simplifying the analysis. We present the main ideas of the proof techniques in Section 5 and provide the detailed proof in Appendix B and Appendix C.

Combining Proposition 4 with Theorem 2, we obtain our main result: a bound on the expected TS regret in $O(d/\alpha\sqrt{T\log(\beta T/d)})$. To the best of our knowledge, this is the first regret bound for *any logistic bandit algorithm* that does not scale exponentially with the logistic function's parameter $\beta$ or with the problem dimension $d$ and remains independent of the number of actions.

**Theorem 5** *For any $\beta > 0$, and for all $\mathcal{A}, \mathcal{O} \subseteq \mathbf{B}_d(0,1)$ with minimax alignment constant $\alpha > 0$, under the logistic bandit setting with logistic function $\phi_\beta(x)$, the TS regret is bounded as*

$$\mathbb{E}[\mathrm{Regret}(T)] \leqslant \frac{4d}{\alpha}\sqrt{T\log\left(3 + 6\beta\sqrt{\frac{T}{2d}}\right)}.$$

**Proof 6** *Combining Theorem 2 with Proposition 4, we upper bound the entropy $\mathrm{H}(\Theta_\varepsilon)$ by the cardinality of the $\varepsilon$-net to get a regret bound of $4\alpha^{-1}\sqrt{dT\left(\log(|\Theta_\varepsilon|) + \frac{1}{2}\varepsilon^2\beta^2 T\right)}$. To define $\Theta_\varepsilon$, we set $\mathcal{O}_\varepsilon$ as the $\varepsilon$-net of smallest cardinality. As the parameter space $\mathcal{O}$ is within the Euclidean unit ball, we use Lemma 23 to control the covering number, $\log(|\Theta_\varepsilon|) \leqslant d\log(1 + 2/\varepsilon)$, and upper bound the TS regret as*

$$\mathbb{E}[\mathrm{Regret}(T)] \leqslant 4/\alpha\sqrt{dT\left(d\log\left(1 + \frac{2}{\varepsilon}\right) + \frac{1}{2}\varepsilon^2\beta^2 T\right)}.$$

*Finally, setting $\varepsilon = \sqrt{2d}/\sqrt{\beta^2 T}$ and rearranging terms inside the logarithm yields the desired result.*

Remarkably, our result does not depend on the fragility dimension $\eta$. This is important as, except in the case where $\alpha = 1$, the fragility dimension can grow exponentially with the problem dimension $d$ (see Dong et al. (2019, Remark 3)). In general, the dependence on $\alpha$ cannot be removed; as shown by Dong et al. (2019, Proposition 11), there cannot be an $\alpha$-independent upper bound that is polynomial in $d$ and sublinear in $T$.

Nevertheless, the following corollaries identify specific settings where this dependence on the minimax alignment constant $\alpha$ disappears. We illustrate the improvement of Corollary 7 over previous works through numerical experiments on a synthetic logistic bandit problem. The results are presented in Section 6.

**Corollary 7** *For any $\beta$, under the logistic bandit setting with logistic function $\phi_\beta(x)$, let $\mathcal{A} \subseteq \mathbf{B}_d(0,1)$ and $\mathcal{O} \subseteq \mathbf{S}_d(0,1)$ be such that $\mathcal{O} \subseteq \mathcal{A}$. Then the TS regret is bounded as*

$$\mathbb{E}[\mathrm{Regret}(T)] \leqslant 2d\sqrt{T\log\left(3 + 6\beta\sqrt{\frac{T}{2d}}\right)}.$$

**Proof 8** *If $\mathcal{O} \subseteq \mathbf{S}_d(0,1)$ and if $\mathcal{O} \subseteq \mathcal{A}$, then for each $\theta \in \mathcal{O}$, there exists an action $a \in \mathcal{A}$ such that $a = \theta$ and $\langle a, \theta \rangle = 1$, implying $\alpha = 1$. In this setting, we can use a tighter bound on the information, $\Gamma_t \leqslant 4d/\alpha^2$, presented in Section 5. Using this result together with Theorem 2 as in Theorem 5 concludes the proof.*

**Corollary 9** *For any $\beta$, under the logistic bandit setting with logistic function $\phi_\beta(x)$, there exists an action space $\mathcal{A}$ with $|\mathcal{A}| \leqslant 2d \cdot 3^{d-1}$ such that for any $\mathcal{O} \subseteq \mathbf{S}_d(0,1)$, the TS regret is bounded as*

$$\mathbb{E}[\mathrm{Regret}(T)] \leqslant 8d \sqrt{T \log \left( 3 + 6\beta \sqrt{\frac{T}{2d}} \right)}.$$

**Proof 10** *Starting from Theorem 5, we have to construct $\mathcal{A}$ such that its minimax alignment constant $\alpha$ is greater than or equal to $\frac{1}{2}$ for any $\mathcal{O} \subseteq \mathbf{S}_d(0,1)$. This is satisfied if $\mathcal{A}$ is a $\frac{1}{2}$-net for $\mathbf{S}_d(0,1)$. Setting $\mathcal{A}$ as the $\frac{1}{2}$-net of minimal cardinality, from Lemma 24, we have $|\mathcal{A}| \leqslant 2d \cdot 3^{d-1}$.*

In the above corollaries, the regret bound matches the $\Omega(d\sqrt{T})$ minimax lower bound for linear bandits of Dani et al. (2008), up to logarithmic factors. Moreover, its dependence on $d$ and $\sqrt{T}$ aligns with the instance-dependent lower bound of Abeille et al. (2021), implying that the bound is tight in both $d$ and $T$ up to logarithmic and problem-dependent constants.

## 5 Analysis

This section presents the key ideas underlying the proof of our main proposition, Proposition 4. For simplicity and clarity of exposition, we focus first on the specific setting of Corollary 7, which corresponds to the case where $\alpha = 1$. The generalization of these arguments to arbitrary parameter and action spaces follows the same reasoning but requires additional notation and handling the misalignment between the parameter and action spaces. This extension is deferred to Appendix C.

The proof is organized into three parts: Section 5.1 presents a lower bound on the mutual information via the variance of reward probability; Section 5.2 proves an upper bound on the squared expected regret using a surrogate logistic function; and finally Section 5.3 explains how we derive a general upper bound on the ratio of expected variances and use the limit case $\beta \to \infty$ to prove a uniform upper bound over $\beta > 0$.

To lighten notation, we omit the subscript $t$ for the remainder of the section. We recall that under the logistic bandit setting with logistic function $\phi_\beta$, the reward $R(\hat{A}, \Theta)$ is sampled according to a Bernoulli distribution with probability $\phi_\beta(\langle \hat{A}, \Theta \rangle)$. We introduce the notation $\mathrm{Bern}(\phi_\beta(\langle \hat{A}, \Theta \rangle))$ to make the setting more explicit. With this notation, the information ratio can be written as

$$\Gamma = \frac{\mathbb{E}[\mathrm{Bern}(\phi_\beta(\langle A^\star, \Theta \rangle)) - \mathrm{Bern}(\phi_\beta(\langle \hat{A}, \Theta \rangle))]^2}{\mathrm{I}(\Theta; \mathrm{Bern}(\phi_\beta(\langle \hat{A}, \Theta \rangle)), \hat{\Theta})}.$$

### 5.1 Lower bounding the mutual information via the variance of reward probability

We start by stating a useful lemma that relates the variance of a $[0, 1]$ random variable $U$ to the mutual information between $U$ and a Bernoulli outcome with probability $U$. The proof is presented in Appendix A.

**Lemma 11** *Let $U$ be a random variable taking values in $[0, 1]$ and $\mathrm{Bern}(U)$ be a Bernoulli random variable with probability $U$. Then it holds that*

$$\mathrm{I}(U; \mathrm{Bern}(U)) \geqslant 2\mathbb{V}(U).$$

Using Lemma 11, we prove in Lemma 12, that the information that the agent gained about the parameter, $\mathrm{I}(\Theta; \mathrm{Bern}(\phi_\beta(\langle \hat{A}, \Theta \rangle)), \hat{\Theta})$, is at least twice the expected variance of reward probability $\mathbb{E}[\mathbb{V}[\phi_\beta(\langle \hat{A}, \Theta \rangle)|\hat{\Theta}]]$.

**Lemma 12** *Let the logistic function be $\phi_\beta(x)$, then, for Thompson Sampling, it holds that*

$$\mathrm{I}(\Theta; \mathrm{Bern}(\phi_\beta(\langle \hat{A}, \Theta \rangle)), \hat{\Theta}) \geqslant 2\mathbb{E}\left[ \mathbb{V}\left[ \phi_\beta(\langle \hat{A}, \Theta \rangle) \mid \hat{\Theta} \right] \right].$$

**Proof 13** *We start by applying the chain rule. It follows that*

$$
\mathrm{I}(\Theta; \mathrm{Bern}(\phi_\beta(\langle \hat{A}, \Theta \rangle)), \hat{\Theta}) \overset{(i)}{=} \mathrm{I}(\Theta; \hat{\Theta}) + \mathrm{I}(\Theta; \mathrm{Bern}(\phi_\beta(\langle \hat{A}, \Theta \rangle)) \mid \hat{\Theta})
$$
$$
\overset{(j)}{=} \mathrm{I}(\Theta; \mathrm{Bern}(\phi_\beta(\langle \hat{A}, \Theta \rangle)) \mid \hat{\Theta})
$$
$$
\overset{(k)}{\geqslant} \mathbb{E}[\mathrm{I}(\phi_\beta(\langle \pi^\star(\hat{\Theta}), \Theta \rangle); \mathrm{Bern}(\phi_\beta(\langle \pi^\star(\hat{\Theta}), \Theta \rangle))) \mid \hat{\Theta}],
$$

*where (i) follows from the chain-rule; (j) follows as $\Theta$ and $\hat{\Theta}$ are independent conditioned on the history; and (k) follows from conditioning on $\hat{\Theta}$ and applying the data processing inequality. Finally, applying Lemma 11 with $\phi_\beta(\langle \pi^\star(\hat{\theta}), \Theta \rangle)$ in place of the random variable $U$ yields the desired result.*

Intuitively, if the expected variance of reward probability is high, the agent is still exploring new actions and gathering information about the parameter $\Theta$. In such cases, although the instantaneous regret may remain large, the agent continues to learn about the environment, which can allow the information ratio to be controlled. If the expected variance of reward probability is low, we need to show that the agent has identified near-optimal actions and is now incuring small instantaneous regrets. Thus the next part of the proof, Section 5.2, will be focused on making this intuition rigourous by relating the expected instantaneous regret to the expected variance of reward probability.

## 5.2 Upper bounding the squared expected regret using the expected regret surrogate function

This part builds on and improves upon the analysis techniques of Dong et al. (2019, Theorem 5) and similar to them, the following two lemmata will be important to our proof. We restate their proofs in Appendix A.

**Lemma 14** *Let $U, V$ be random vectors in $\mathbb{R}^d$, and let $\tilde{U}, \tilde{V}$ be independent random variables with distributions equal respectively to the marginals of $U, V$, then it holds that*

$$
\mathbb{E}\left[U^\top V\right]^2 \leqslant d \cdot \mathbb{E}\left[\left(\tilde{U}^\top \tilde{V}\right)^2\right].
$$

**Lemma 15** *Let $f : \mathbb{R}_+ \to \mathbb{R}_+$ be such that $f(0) \geqslant 0$ and $f(\zeta)/\zeta$ is non-decreasing over $\zeta \geqslant 0$. Then, for any non-negative random variable $U$, it holds that*

$$
\frac{\mathbb{E}[f(U)]^2}{\mathbb{E}[U]^2} \leqslant \frac{\mathbb{V}[f(U)]}{\mathbb{V}[U]}.
$$

The two lemmata above will be key to relating the expected instantaneous regret to the expected variance of the reward probability. After conditioning on the sampled Thompson Sampling parameter $\hat{\Theta}$, we will express the expected regret as a function of $\hat{\Theta}$ and apply Lemma 15 to relate it to the regret variance. We will then use Lemma 14 to decouple inner-product terms and reveal the dependence on the dimension $d$. Before doing so, however, we introduce additional notation and a few reformulations to make Lemma 15 applicable.

Under the assumptions of Corollary 7, for each $\theta \in \mathcal{O}$, there exists an action $a \in \mathcal{A}$ such that $\langle a, \theta \rangle = 1$. This implies that the expected regret can be written as $\phi_\beta(\langle A^\star, \Theta \rangle) - \phi_\beta(\langle \hat{A}, \Theta \rangle) = \phi_\beta(1) - \phi_\beta(\langle \hat{A}, \Theta \rangle)$. As this quantity will come up often in the analysis, we introduce the *expected regret function* $\Delta_\beta(x) := \phi_\beta(1) - \phi_\beta(1 - x)$. We can then write the expected regret as a function of the difference between the optimal and achieved inner products, as $\Delta_\beta(\langle A^\star, \Theta \rangle - \langle \hat{A}, \Theta \rangle) = \phi_\beta(\langle A^\star, \Theta \rangle) - \phi_\beta(\langle \hat{A}, \Theta \rangle)$.

Importantly, the expected regret function $\Delta_\beta(x)$ applied to the difference of inner products $\langle A^\star, \Theta \rangle - \langle \hat{A}, \Theta \rangle$ maps the interval $[0, 2]$ to $[0, 1]$, and respects $\Delta_\beta(0) = \phi_\beta(1) - \phi_\beta(1 - 0) = 0$, thus meeting the first two conditions from Lemma 15. However, it does not satisfy the third condition, as $\Delta_\beta(x)/x$ first increases, then reaches a maximum between 1 and 2, and finally decreases (see Appendix D and Figure 5). To address this issue, similarly to Dong et al. (2019, B.2. Proof of (18)), we introduce a modified function, referred to as the *expected regret surrogate*, which serves as the tightest upper bound on $\Delta_\beta(x)$ satisfying the last requirement from Lemma 15. The functions $\Delta_\beta$ and $\bar{\Delta}_\beta$ are illustrated in Figure 1.

**Definition 16 (Expected regret surrogate)** *We construct the* expected regret surrogate *function* $\bar{\Delta}_\beta(x)$ *as the tightest upper bound on $\Delta_\beta(x)$ such that $\bar{\Delta}_\beta(x)/x$ is non-decreasing over $x \geqslant 0$.*

*Namely, let $\delta_\beta = \arg\max_{x \in [0,2]} \frac{\Delta_\beta(x)}{x}$, we define $\bar{\Delta}_\beta$ as*

$$\bar{\Delta}_\beta(x) = \begin{cases} \Delta_\beta(x) & x \in [0, \delta_\beta] \\ \Delta_\beta(\delta_\beta) + (x - \delta_\beta) \cdot \Delta'_\beta(\delta_\beta) & x \in \,]\delta_\beta, 2] \end{cases},$$

*where we used that $\Delta'_\beta(\delta_\beta) = \Delta_\beta(\delta_\beta)/\delta_\beta$.*

We are now equipped to state and prove an upper bound on the squared expected regret using the surrogate function.

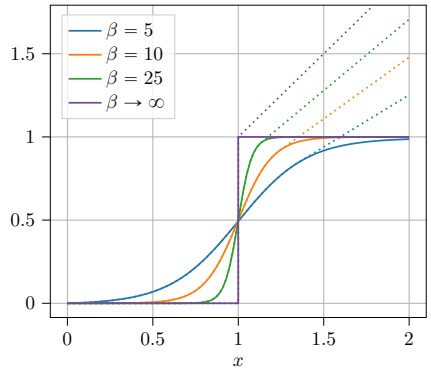

Figure 1: Illustration of the function $\Delta_\beta$ (in solid line) and the function $\bar{\Delta}_\beta$ (in dotted line) for different values of $\beta$.

Figure 2: Illustration of the function $f \circ \Delta_\beta$ (in solid line) and the function $G_\beta$ (in dotted line) for different values of $\beta$ used in the proof of Lemma 19.

**Lemma 17** *Let the expected regret surrogate be defined as in Definition 16. Then, it holds that*

$$\mathbb{E}[\mathrm{Bern}(\phi_\beta(\langle A^\star, \Theta \rangle)) - \mathrm{Bern}(\phi_\beta(\langle \hat{A}, \Theta \rangle))]^2 \leqslant 2d \cdot \mathbb{E}\left[ \mathbb{V}\left[ \bar{\Delta}_\beta \left( 1 - \langle \hat{A}, \Theta \rangle \right) \mid \hat{\Theta} \right] \right].$$

**Proof 18** *Integrating over the Bernoulli outcome, the squared expected regret can be expressed as $\mathbb{E}[(\phi_\beta(\langle A^\star, \Theta \rangle) - \phi_\beta(\langle \hat{A}, \Theta \rangle))]^2 = \mathbb{E}[\Delta_\beta(1 - \langle \hat{A}, \Theta \rangle)]^2$. Since by definition $\bar{\Delta}_\beta(x) \geqslant \Delta_\beta(x) \geqslant 0$, we have $\mathbb{E}[\Delta_\beta(1 - \langle \hat{A}, \Theta \rangle)]^2 \leqslant \mathbb{E}[\mathbb{E}[\bar{\Delta}_\beta(1 - \langle \hat{A}, \Theta \rangle)|\hat{\Theta}]]^2$. We now apply Lemma 15 and obtain*

$$\mathbb{E}[\mathbb{E}[\bar{\Delta}_\beta(1 - \langle \hat{A}, \Theta \rangle)|\hat{\Theta}]]^2 \leqslant \mathbb{E}\left[ \underbrace{\sqrt{\frac{\mathbb{V}\left[ \bar{\Delta}_\beta\left(1 - \langle \hat{A}, \Theta \rangle\right) \mid \hat{\Theta} \right]}{\mathbb{V}\left[ 1 - \langle \hat{A}, \Theta \rangle \mid \hat{\Theta} \right]}}}_{:=U(\hat{\Theta})} \mathbb{E}\left[ 1 - \langle \hat{A}, \Theta \rangle \mid \hat{\Theta} \right] \right]^2$$

$$= \mathbb{E}\left[ U(\hat{\Theta})(\langle \hat{A}, \hat{\Theta} \rangle - \langle A^\star, \hat{\Theta} \rangle) \right]^2 = \mathbb{E}\left[ \langle U(\hat{\Theta})\hat{A}, \Theta - \hat{\Theta} \rangle \right]^2.$$

*We continue by applying Lemma 14 with $U = U(\hat{\Theta})\hat{A}$ and $V = \Theta - \hat{\Theta}$ and get*

$$\mathbb{E}\left[ \langle U(\hat{\Theta})\hat{A}, \Theta - \hat{\Theta} \rangle \right]^2 \leqslant d \cdot \mathbb{E}\left[ \langle U(\hat{\Theta})\hat{A}, \Theta - \tilde{\Theta} \rangle^2 \right] = d \cdot \mathbb{E}\left[ U(\hat{\Theta})^2 \mathbb{E}\left[ \langle \hat{A}, \Theta - \tilde{\Theta} \rangle^2 | \hat{\Theta} \right] \right]$$

$$\overset{(i)}{=} d \cdot \mathbb{E}\left[ \frac{\mathbb{V}\left[ \bar{\Delta}_\beta\left(1 - \langle \hat{A}, \Theta \rangle\right) \mid \hat{\Theta} \right]}{\mathbb{V}\left[ 1 - \langle \hat{A}, \Theta \rangle \mid \hat{\Theta} \right]} 2\mathbb{V}\left[ \langle \hat{A}, \Theta \rangle \mid \hat{\Theta} \right] \right]$$

$$= 2d \cdot \mathbb{E}\left[ \mathbb{V}\left[ \bar{\Delta}_\beta\left(1 - \langle \hat{A}, \Theta \rangle\right) \mid \hat{\Theta} \right] \right],$$

*where (i) follows as*

$$\mathbb{E}\left[\langle\hat{A},\Theta-\tilde{\Theta}\rangle^2|\hat{\Theta}\right] = \mathbb{E}[\langle\hat{A},\Theta\rangle^2|\hat{\Theta}] - 2\mathbb{E}[\langle\hat{A},\Theta\rangle|\hat{\Theta}]\mathbb{E}[\langle\hat{A},\tilde{\Theta}\rangle|\hat{\Theta}] + \mathbb{E}[\langle\hat{A},\tilde{\Theta}\rangle^2|\hat{\Theta}]$$
$$= 2\mathbb{E}[\langle\hat{A},\Theta\rangle^2|\hat{\Theta}] - 2\mathbb{E}[\langle\hat{A},\Theta\rangle|\hat{\Theta}]^2 = 2\mathbb{V}[\langle\hat{A},\Theta\rangle\mid\hat{\Theta}].$$

*Finally rearranging the terms gives the claimed result.*

Combining Lemma 12 and Lemma 17, we get that the information ratio $\Gamma$ is bounded by

$$\Gamma \leqslant d\cdot\frac{\mathbb{E}\left[\mathbb{V}\left[\bar{\Delta}_\beta\left(1-\langle\hat{A},\Theta\rangle\right)\mid\hat{\Theta}\right]\right]}{\mathbb{E}\left[\mathbb{V}\left[\Delta_\beta\left(1-\langle\hat{A},\Theta\rangle\right)\mid\hat{\Theta}\right]\right]},$$

where we use that $\mathbb{V}\left[\phi_\beta(\langle\hat{A},\Theta\rangle)\mid\hat{\Theta}\right] = \mathbb{V}\left[\Delta_\beta(1-\langle\hat{A},\Theta\rangle)\mid\hat{\Theta}\right]$ by definition of $\Delta_\beta$. We are close to obtaining a bound on the TS information ratio that is linear in $d$ and independent of $\beta$ and of the cardinality of the action and parameter spaces. We still have to control the ratio of expected variances between $\bar{\Delta}_\beta$ and $\Delta_\beta$, as we could not apply Lemma 15 directly on $\Delta_\beta$ and had to work instead with its surrogate $\bar{\Delta}_\beta$.

## 5.3 Bounding the ratio of expected variances over the functions $\bar{\Delta}_\beta$ and $\Delta_\beta$

The last part of the proof takes care of controlling the ratio of expected variances between the regret probability function $\Delta_\beta$ and expected regret surrogate $\bar{\Delta}_\beta$. The full proof is presented in Appendix B.

**Lemma 19** *Let $\Delta_\beta(x) = \phi_\beta(1) - \phi_\beta(1-x)$ and its surrogate $\bar{\Delta}_\beta$ as in Definition 16. Then, for all $\beta > 0$, it holds that*

$$\frac{\mathbb{E}\left[\mathbb{V}\left[\bar{\Delta}_\beta\left(1-\langle\hat{A},\Theta\rangle\right)\mid\hat{\Theta}\right]\right]}{\mathbb{E}\left[\mathbb{V}\left[\Delta_\beta\left(1-\langle\hat{A},\Theta\rangle\right)\mid\hat{\Theta}\right]\right]} \leqslant 4.$$

**Sketch of proof** *The proof starts by observing that the functions $\Delta_\beta$ and $\bar{\Delta}_\beta$ are equal on the interval $[0,\delta_\beta]$, and then diverge at most at a rate $\Delta'_\beta(\delta_\beta)$. To control this divergence, we introduce the function $G_\beta(x) = (x-\delta_\beta)\Delta'_\beta(\delta_\beta)$ for $x\in[\delta_\beta,2]$, and is equal to zero over $[0,\delta_\beta]$. We show that*

$$\frac{\mathbb{E}\left[\mathbb{V}\left[\bar{\Delta}_\beta\left(1-\langle\hat{A},\Theta\rangle\right)\mid\hat{\Theta}\right]\right]}{\mathbb{E}\left[\mathbb{V}\left[\Delta_\beta\left(1-\langle\hat{A},\Theta\rangle\right)\mid\hat{\Theta}\right]\right]} \leqslant \left(1+\sqrt{\frac{\mathbb{E}\left[\mathbb{V}\left[G_\beta\left(1-\langle\hat{A},\Theta\rangle\right)\mid\hat{\Theta}\right]\right]}{\mathbb{E}\left[\mathbb{V}\left[\Delta_\beta\left(1-\langle\hat{A},\Theta\rangle\right)\mid\hat{\Theta}\right]\right]}}\right)^2.$$

*Next, we apply a cropping function $f_\beta(x) = \min(\Delta_\beta(\delta_\beta),x)$ to the function $\Delta_\beta(x)$, contracting the squared difference $(\Delta_\beta(x) - \Delta_\beta(y))^2$ for every $x,y\in[0,2]$, and thus reducing the expected variance. The functions $f_\beta\circ\Delta_\beta$ and $G_\beta$ are illustrated in Figure 2. After expressing the expected variances as the expected squared difference over the variable $1-\langle\pi_\star(\hat{\Theta}),\Theta\rangle$ and $1-\langle\pi_\star(\hat{\Theta}),\tilde{\Theta}\rangle$ where $\hat{\Theta},\Theta,\tilde{\Theta}$ are i.i.d., we consider any triplet $\theta_0,\theta_1,\theta_2\in\mathcal{O}$ and bound the worst-case ratio for $\hat{\Theta},\Theta,\tilde{\Theta}\in\{\theta_0,\theta_1,\theta_2\}$. Maximizing the ratio can be reduced as a convex optimization problem, and we get that the ratio of expected variances between $G_\beta$ and $f_\beta\circ\Delta_\beta$ is less than or equal to $(2-\delta_\beta)^2/\delta_\beta^2$. This ratio can be controlled by the asymptotic case $\beta\to\infty$ and is smaller than or equal to 4 (see Appendix D). Finally, plugging this result into the above inequality gives the claimed result.*

## 6 Numerical Experiments

To illustrate the improvement of our regret analysis compared to previous works, we perform numerical experiments on a synthetic problem. We consider a logistic bandit problem in dimension $d = 10$, with time horizon $T = 200$, and with slope parameter $\beta$ ranging from 0.25 to 10. For both action space and parameter space, we use the closed $d$-dimensional unit sphere, $\mathcal{A} = \mathcal{O} = \mathbf{S}_d(0,1)$ and assume a uniform prior distribution for the parameter $\Theta$. We compute the expected regret of the Thompson Sampling algorithm using an MCMC method and compare it to three Bayesian regret bounds that hold for continuous spaces: our Corollary 7, Russo & Van Roy (2014, Proposition 10) , and Dong et al. (2019, Proposition 17) adapted to be compatible with Dong & Van Roy (2018, Theorem 1) (see Appendix E).

The results are presented in Figure 3. The left sub-figure shows the evolution of the expected regret and the regret bounds for two different logistic function values, $\beta \in \{2, 4\}$. For both values, our bound remains tighter across the entire horizon and is less sensitive to increasing $\beta$. The right sub-figure compares the different regret bounds at $T = 200$ for $\beta \in [0.25, 10]$. We observe that our bound is competitive across the whole range and quickly becomes orders of magnitude tighter. Importantly, we observe that while our bound increases only logarithmically, both Russo & Van Roy (2014, Proposition 10) and Dong et al. (2019, Proposition 17) grow exponentially with $\beta$ and quickly become vacuous. We note that the expected regret decreases for larger $\beta$. This was anticipated since, for large values of $\beta$, the distinction between near-optimal and suboptimal actions becomes more pronounced, facilitating the identification of near-optimal actions.

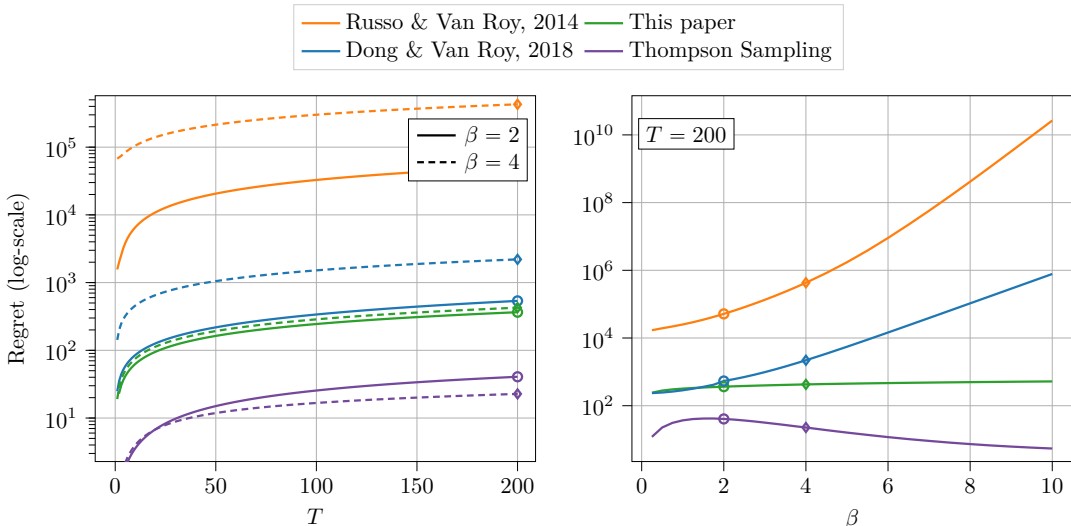

Figure 3: The left sub-figure compares the evolution of the bounds and the expected regret with the time steps $T$ for $\beta \in \{2, 4\}$. The right sub-figure illustrates the behavior of the bounds and the expected regret at time $T = 200$ for values of $\beta$ ranging in $[0.25, 10]$.

## 7 Conclusion and Future Work

In this work, we analyze the performance of the Thompson Sampling algorithm for logistic bandit problems. In this setting, the agent sequentially selects actions $a \in \mathcal{A} \subset \mathbb{R}^d$ and receives binary rewards with probability given by a logistic function $\exp(\beta\langle a, \theta\rangle)/(1+\exp(\beta\langle a, \theta\rangle))$, with slope parameter $\beta$ and an unknown parameter $\theta \in \mathcal{O} \subset \mathbb{R}^d$. Building on the information-theoretic framework from Russo & Van Roy (2016), we study the information ratio, a key statistic that captures the trade-off between exploration and exploitation in bandit problems. We show that the information ratio of Thompson Sampling for logistic bandits can be bounded using only the dimension of the problem, $d$, and $\alpha$, the minimax constant measuring the alignment between the action and parameter spaces. Importantly, our bound is independent of the slope parameter $\beta$ and of the cardinality of the action and parameter spaces.

Using this result, we prove a regret bound of $O(d/\alpha\sqrt{T\log(\beta T/d)})$, which scales only logarithmically with $\beta$, representing a significant improvement over prior works, all of which scale either exponentially with $\beta > 0$ or depend on the cardinality of the action set. To the best of our knowledge, this is the first regret bound for *any logistic bandit algorithm* that does not scale exponentially with $\beta$ while remaining independent of the action set's cardinality. Finally, we present specific settings where the dependence on $\alpha$ can be controlled. For instance, when the action space fully encompasses the parameter space, the regret of Thompson Sampling scales as $\tilde{O}(d\sqrt{T})$. Overall, these results help to explain the empirical performance of Thompson Sampling in logistic bandit problems across different logistic regimes and for large or continuous action spaces, and highlight the importance of alignment between the action and parameter spaces.

An exciting direction for future work is to extend our analysis to the broader class of generalized linear bandits. The properties of the logistic function that we leverage in our analysis could be shared by other classes of link functions and could be used to derive regret bounds using a similar analysis of the information ratio as we performed in Section 5.

Another interesting research direction is to use the result in this paper to derive regret bounds for logistic bandits in the frequentist setting. A promising way is to apply our information-theoretic analysis to the optimistic information directed sampling algorithm introduced by Neu et al. (2024). We believe that this approach could lead to new and improved frequentist bounds for logistic bandits.

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

## Appendix

The appendix is organized as follows:

- Appendix A introduces four important lemmata for our main results;

- Appendix B formalizes the proof for controlling the ratio of expected variances between the functions $\bar{\Delta}_\beta$ and $\Delta_\beta$;

- Appendix C extends our information ratio analysis to general action and parameter spaces;

- Appendix D provides an analysis of the functions $\delta_\beta$ and $\Delta_\beta(\delta_\beta)$;

- Appendix E illustrates how translating directly linear bandits bounds to the logistic bandit setting leads to regret bounds scaling exponentially with $\beta$;

- Appendix F elaborates on the gaps in the previous literature mentioned in Section 1;

- Appendix G provides additional details on the numerical experiments presented in Section 6.

## A   Useful lemmata

**Lemma 11** *Let $U$ be a random variable taking values in $[0, 1]$ and $\mathrm{Bern}(U)$ be a Bernoulli random variable with probability $U$. Then it holds that*

$$I(U; \mathrm{Bern}(U)) \geqslant 2\mathbb{V}(U).$$

**Proof 20** *Using Polyanskiy & Wu (2025, Theorem 3.4.d), we decompose the mutual information between $U$ and $\mathrm{Bern}(U)$ as*

$$I(U; \mathrm{Bern}(U)) = h(\mathrm{Bern}(U)) - h(\mathrm{Bern}(U)|U).$$

*Following Duchi (2019, Example 2.2) notation, we define $h_2(p) := -p\log(p) - (1-p)\log(1-p)$ for $p \in [0, 1]$ and rewrite the mutual information as*

$$I(U; \mathrm{Bern}(U)) = h_2(\mathbb{E}[U]) - \mathbb{E}[h_2(U)]. \tag{3}$$

*From a Taylor expansion of $h_2(x)$ we have that $h_2(x) = h_2(p) + (x-p)h_2'(p) + \frac{1}{2}(x-p)^2 h_2''(\xi)$, for some $\xi \in (0, 1)$ as $h_2''$ is continuous on the interval $[0, 1]$. We compute the second derivative of $h_2$ and get $h_2''(\xi) = -\frac{1}{\xi(1-\xi)}$ for $\xi \in (0, 1)$. This function is concave and maximal at $\xi = 1/2$, where it takes the value $h_2''(1/2) = -4$. We then have that for all $x \in [0, 1]$ and all $p \in [0, 1]$,*

$$h_2(x) \leqslant h_2(p) + (x-p)h_2'(p) - 2(x-p)^2.$$

*Using this fact for $x = U$ and $p = \mathbb{E}[U]$, we have that*

$$h_2(U) \leqslant h_2(\mathbb{E}[U]) + (U - \mathbb{E}[U])h_2'(\mathbb{E}[U]) - 2(U - \mathbb{E}[U])^2.$$

*Applying the last inequality to the second term in eq. (3), it comes that*

$$I(U; \mathrm{Bern}(U)) \geqslant \mathbb{E}\left[h_2(\mathbb{E}[U]) - h_2(\mathbb{E}[U]) - (U - \mathbb{E}[U])h_2'(\mathbb{E}[U]) + 2(U - \mathbb{E}[U])^2\right].$$

*Finally, simplifying terms and taking the expectation gives the desired result.*

**Lemma 14** *Let $U, V$ be random vectors in $\mathbb{R}^d$, and let $\tilde{U}, \tilde{V}$ be independent random variables with distributions equal respectively to the marginals of $U, V$, then it holds that*

$$\mathbb{E}\left[U^\top V\right]^2 \leqslant d \cdot \mathbb{E}\left[\left(\tilde{U}^\top \tilde{V}\right)^2\right].$$

**Proof 21** *This useful fact is presented in Dong et al. (2019, Lemma 16). We provide the full proof below.*

*Let $Q = \mathbb{E}[VV^\top]$, then*

$$\mathbb{E}[U^\top V]^2 = \mathbb{E}[U^\top Q^{1/2}Q^{-1/2}V] = \mathbb{E}[(Q^{1/2}U)^\top(Q^{-1/2}V)]^2$$

$$\overset{(i)}{\leqslant} \mathbb{E}\left[\sqrt{(Q^{1/2}U)^\top(Q^{1/2}U)}\sqrt{(Q^{-1/2}V)^\top(Q^{-1/2}V)}\right]^2$$

$$\overset{(j)}{\leqslant} \mathbb{E}[(Q^{1/2}U)^\top(Q^{1/2}U)]\mathbb{E}[(Q^{-1/2}V)^\top(Q^{-1/2}V)] = \mathbb{E}[U^\top QU]\mathbb{E}[V^\top Q^{-1}V]$$

$$\overset{(k)}{=} \mathbb{E}[U^\top \mathbb{E}[VV^\top]U]\left(\text{Tr}(Q^{-1}\text{Cov}(V)) + \mathbb{E}[V]^\top Q^{-1}\mathbb{E}[V]\right)$$

$$= \mathbb{E}[\tilde{U}^\top \tilde{V}\tilde{V}^\top \tilde{U}]\left(\text{Tr}(Q^{-1}(\mathbb{E}[VV^\top] - \mathbb{E}[V]\mathbb{E}[V]^\top)) + (Q^{-1/2}\mathbb{E}[V])^\top(Q^{-1/2}\mathbb{E}[V])\right)$$

$$= \mathbb{E}[(\tilde{U}^\top \tilde{V})^2]\left(\text{Tr}(\mathbb{I}_d) - \text{Tr}(Q^{-1}\mathbb{E}[V]\mathbb{E}[V]^\top) + \text{Tr}((Q^{-1/2}\mathbb{E}[V])(Q^{-1/2}\mathbb{E}[V])^\top)\right)$$

$$= \mathbb{E}[(\tilde{U}^\top \tilde{V})^2]\left(d - \text{Tr}(Q^{-1/2}\mathbb{E}[V]\mathbb{E}[V]^\top Q^{-1/2}) + \text{Tr}(Q^{-1/2}\mathbb{E}[V]\mathbb{E}[V]^\top Q^{-1/2})\right)$$

$$= d \cdot \mathbb{E}[(\tilde{U}^\top \tilde{V})^2],$$

*where (i) follows from the Cauchy-Schwarz inequality applied to the inner product; (j) follows from the Cauchy-Schwarz inequality applied to the expectation; (k) follows from the identity that for any random vector $W$ and deterministic matrix $M$, we have $\mathbb{E}[W^\top MW] = \text{Tr}(M, \text{Cov}(W)) + \mathbb{E}[W]^\top M, \mathbb{E}[W]$. The remainder of the proof follows by expanding the covariance as $\text{Cov}(V) = \mathbb{E}[VV^\top] - \mathbb{E}[V]\mathbb{E}[V]^\top$, rewriting the inner product as the trace of an outer product, and using the fact that the trace of a product of symmetric matrices is invariant under cyclic permutations.*

**Lemma 15** *Let $f : \mathbb{R}_+ \to \mathbb{R}_+$ be such that $f(0) \geqslant 0$ and $f(\zeta)/\zeta$ is non-decreasing over $\zeta \geqslant 0$. Then, for any non-negative random variable $U$, it holds that*

$$\frac{\mathbb{E}[f(U)]^2}{\mathbb{E}[U]^2} \leqslant \frac{\mathbb{V}[f(U)]}{\mathbb{V}[U]}.$$

**Proof 22** *This result is presented in Dong et al. (2019, Lemma 18). We show that it can be obtained by combining Cauchy-Schwarz inequality and Chebyshev's association inequality.*

*Let $g(\zeta) = f(\zeta)/\zeta$ for $\zeta > 0$, and set $g(0) := \lim_{\zeta\downarrow0} f(\zeta)/\zeta$. By assumption, $g$ is non-negative and non-decreasing. We have that*

$$\frac{\mathbb{E}[f(U)]^2}{\mathbb{E}[U]^2} = \frac{\mathbb{E}[g(U)U^2]}{\mathbb{E}[U]^2} \overset{(i)}{\leqslant} \frac{\mathbb{E}[g(U)^2U]\mathbb{E}[U]}{\mathbb{E}[U]^2} = \frac{\mathbb{E}[g(U)^2U]}{\mathbb{E}[U]} \overset{(j)}{\leqslant} \frac{\mathbb{E}[g(U)^2U^2]}{\mathbb{E}[U^2]} = \frac{\mathbb{E}[f(U)^2]}{\mathbb{E}[U^2]},$$

*where (i) follows Cauchy-Schwarz inequality $|\mathbb{E}[XY]|^2 \leqslant \mathbb{E}[X^2]\mathbb{E}[Y^2]$ applied with $X = g(U)\sqrt{U}$ and $Y = \sqrt{U}$; and (j) follows from Chebyshev's association inequality (Boucheron et al., 2013, Theorem 2.14). We then have that $\mathbb{E}[f(U)]^2\mathbb{E}[U^2] \leqslant \mathbb{E}[f(U)^2]\mathbb{E}[U]^2$. Subtracting $\mathbb{E}[U]^2\mathbb{E}[f(U)]^2$ on both sides we get that*

$$\mathbb{E}[f(U)]^2\left(\mathbb{E}[U^2] - \mathbb{E}[U]^2\right) \leqslant \mathbb{E}[U]^2\left(\mathbb{E}[f(U)^2] - \mathbb{E}[f(U)]^2\right) = \mathbb{E}[f(U)]^2\mathbb{V}[U] \leqslant \mathbb{E}[U]^2\mathbb{V}[f(U)].$$

*Finally, rearranging the terms gives the desired result.*

The following two lemmata are particularly useful to control the covering number in Euclidean balls and spheres.

**Lemma 23 (van Handel (2016, Lemma 5.13))** *Let $\mathbf{B}_d(0,1)$ denote the $d$-dimensional closed Euclidean unit ball. We have $|\mathcal{N}(\mathbf{B}_d(0,1), ||\cdot||_2, \varepsilon)| = 1$ for $\varepsilon \geqslant 1$ and for $0 < \varepsilon < 1$, we have*

$$\left(\frac{1}{\varepsilon}\right)^d \leqslant |\mathcal{N}(\mathbf{B}_d(0,1), ||\cdot||_2, \varepsilon)| \leqslant \left(1 + \frac{2}{\varepsilon}\right)^d.$$

**Lemma 24 (Polyanskiy & Wu (2025, Corollary 27.4))** *Let* $\mathbf{S}_d(0,1)$ *denote the d-dimensional Euclidean unit sphere. We have* $|\mathcal{N}(\mathbf{S}_d(0,1),||\cdot||_2,\varepsilon) = 1$ *for* $\varepsilon \geqslant 1$ *and for* $0 < \varepsilon < 1$*, we have*

$$\left(\frac{1}{2\varepsilon}\right)^{d-1} \leqslant |\mathcal{N}(\mathbf{S}_d(0,1),||\cdot||_2,\varepsilon)| \leqslant 2d\left(1+\frac{1}{\varepsilon}\right)^{d-1}.$$

## B  Bounding the ratio of expected variances over the functions $\bar{\Delta}_\beta$ and $\Delta_\beta$

**Lemma 19** *Let* $\Delta_\beta(x) = \phi_\beta(1) - \phi_\beta(1-x)$ *and its surrogate* $\bar{\Delta}_\beta$ *as in Definition 16. Then, for all* $\beta > 0$*, it holds that*

$$\frac{\mathbb{E}\left[\mathbb{V}\left[\bar{\Delta}_\beta\left(1-\langle\hat{A},\Theta\rangle\right) \mid \hat{\Theta}\right]\right]}{\mathbb{E}\left[\mathbb{V}\left[\Delta_\beta\left(1-\langle\hat{A},\Theta\rangle\right) \mid \hat{\Theta}\right]\right]} \leqslant 4.$$

We begin the proof by noting that the ratio of expected variances can be written as a ratio of expected squared differences,

$$\frac{\mathbb{E}[\mathbb{V}[\bar{\Delta}_\beta(1-\langle\hat{A},\Theta\rangle)|\hat{\Theta}]]}{\mathbb{E}[\mathbb{V}[\Delta_\beta(1-\langle\hat{A},\Theta\rangle)|\hat{\Theta}]]} = \frac{\mathbb{E}[(\bar{\Delta}_\beta(1-\langle\hat{A},\Theta\rangle) - \bar{\Delta}_\beta(1-\langle\hat{A},\tilde{\Theta}\rangle))^2]}{\mathbb{E}[(\Delta_\beta(1-\langle\hat{A},\Theta\rangle) - \Delta_\beta(1-\langle\hat{A},\tilde{\Theta}\rangle))^2]} = \frac{\mathbb{E}[(\bar{\Delta}_\beta(X) - \bar{\Delta}_\beta(\tilde{X}))^2]}{\mathbb{E}[(\Delta_\beta(X) - \Delta_\beta(\tilde{X}))^2]}.$$

where $\tilde{\Theta}$ is a random variable independent and identically distributed as $\Theta$. We can write it in a more compact form using $X = 1 - \langle\hat{A},\Theta\rangle$ and $\tilde{X} = 1 - \langle\hat{A},\tilde{\Theta}\rangle$.

We continue by observing that for $x \in [0,\delta_\beta[$, we have $\bar{\Delta}_\beta(x) - \Delta_\beta(x) = 0$ and for $x \in [\delta_\beta, 2]$, we have that $\bar{\Delta}_\beta(x) - \Delta_\beta(x) \leqslant \Delta_\beta(\delta_\beta)/\delta_\beta(x - \delta_\beta)$ as $\Delta_\beta(\delta_\beta) \leqslant \Delta_\beta(x)$. Thus we can upper bound the difference $\bar{\Delta}_\beta(x) - \Delta_\beta(x)$ by a function $G_\beta(x)$ such that

$$G_\beta(x) = \begin{cases} 0 & x \in [0,\delta_\beta[ \\ (x-\delta_\beta)\Delta_\beta(\delta_\beta)/\delta_\beta & x \in [\delta_\beta, 2]. \end{cases}$$

We note that for all $x,y \in [0,2]$, we have that

$$|\Delta_\beta(x) + G_\beta(x) - \Delta_\beta(y) - G_\beta(y)| \geqslant |(\bar{\Delta}_\beta(x) - (\bar{\Delta}_\beta(y)|.$$

Indeed, for $x,y \in [0,\delta_\beta]$, we have $|\Delta_\beta(x) + G_\beta(x) - \Delta_\beta(y) - G_\beta(y)| = 0 = |(\bar{\Delta}_\beta(x) - \bar{\Delta}_\beta(y))|$; for $x,y \in ]\delta_\beta], 2$, we have $|\Delta_\beta(x) + G_\beta(x) - \Delta_\beta(y) - G_\beta(y)| = |\Delta_\beta(x) - \Delta_\beta(y) + \Delta_\beta(\delta_\beta)/\delta_\beta(x - y)| \geqslant |\Delta_\beta(\delta_\beta) - \Delta_\beta(\delta_\beta) + \Delta_\beta(\delta_\beta)/\delta_\beta(x-y)| = |(\bar{\Delta}_\beta(x) - \bar{\Delta}_\beta(y)|$; and for $x \in ]\delta_\beta], 2, y \in [0,\delta_\beta]$, we have again $|\Delta_\beta(x) + G_\beta(x) - \Delta_\beta(y) - G_\beta(y)| = |\Delta_\beta(x) - \Delta_\beta(y) + \Delta_\beta(\delta_\beta)/\delta_\beta(x - \delta_\beta)| \geqslant |\Delta_\beta(\delta_\beta) - \Delta_\beta(y) + \Delta_\beta(\delta_\beta)/\delta_\beta(x - \delta_\beta)| = |\bar{\Delta}_\beta(x) - \bar{\Delta}_\beta(y)|$. As the inequality $(\Delta_\beta(x) + G_\beta(x) - \Delta_\beta(y) - G_\beta(y))^2 \geqslant (\bar{\Delta}_\beta(x) - \bar{\Delta}_\beta(y))^2$ holds everywhere, it holds also in expectation and we have that

$$\frac{\mathbb{E}[(\bar{\Delta}_\beta(X) - \bar{\Delta}_\beta(\tilde{X}))^2]}{\mathbb{E}[(\Delta_\beta(X) - \Delta_\beta(\tilde{X}))^2]} \leqslant \frac{\mathbb{E}[(\Delta_\beta(X) - \Delta_\beta(\tilde{X}) + G_\beta(X) - G_\beta(\tilde{X}))^2]}{\mathbb{E}[(\Delta_\beta(X) - \Delta_\beta(\tilde{X}))^2]}$$

$$= 1 + 2\frac{\mathbb{E}[(\Delta_\beta(X) - \Delta_\beta(\tilde{X}))(G_\beta(X) - G_\beta(\tilde{X}))]}{\mathbb{E}[(\Delta_\beta(X) - \Delta_\beta(\tilde{X}))^2]} + \frac{\mathbb{E}[(G_\beta(X) - G_\beta(\tilde{X}))^2]}{\mathbb{E}[(\Delta_\beta(X) - \Delta_\beta(\tilde{X}))^2]}$$

$$\leqslant 1 + 2\sqrt{\frac{\mathbb{E}[(G_\beta(X) - G_\beta(\tilde{X}))^2]}{\mathbb{E}[(\Delta_\beta(X) - \Delta_\beta(\tilde{X}))^2]}} + \frac{\mathbb{E}[(G_\beta(X) - G_\beta(\tilde{X}))^2]}{\mathbb{E}[(\Delta_\beta(X) - \Delta_\beta(\tilde{X}))^2]}$$

$$= \left(1 + \sqrt{\frac{\mathbb{E}[(G_\beta(X) - G_\beta(\tilde{X}))^2]}{\mathbb{E}[(\Delta_\beta(X) - \Delta_\beta(\tilde{X}))^2]}}\right)^2, \tag{4}$$

where in the second inequality, we used Cauchy-Schwarz inequality. We can then focus on studying the ratio of expected squared differences between $G_\beta$ and $\Delta_\beta$. We will first apply a transformation $f_\beta(x) = \min(\Delta_\beta(\delta_\beta), \max(\Delta_\beta(1), x))$ to crop the values of $\Delta_\beta$ that are above $\Delta_\beta(\delta_\beta)$. We can write the resulting function $f_\beta \circ \Delta_\beta$ as

$$f_\beta \circ \Delta_\beta(x) = \begin{cases} \Delta_\beta(1) & x \in [0,1] \\ \Delta_\beta(x) & x \in ]1, \delta_\beta[ \\ \Delta_\beta(\delta_\beta) & x \in [\delta_\beta, 2[. \end{cases}$$

We observe that the transformation $f_\beta$ contracts the function $\Delta_\beta$ as, for all $x, y \in [0, 2]$, we have $|f_\beta \circ \Delta_\beta(x) - f_\beta \circ \Delta_\beta(y)| \leqslant |\Delta_\beta(x) - \Delta_\beta(y)|$. As the inequality holds everywhere, it holds also in expectation and we have that $\mathbb{E}[(f_\beta \circ \Delta_\beta(X) - f_\beta \circ \Delta_\beta(\tilde{X}))^2] \leqslant \mathbb{E}[(\Delta_\beta(X) - \Delta_\beta(\tilde{X}))^2]$ and it follows that

$$\frac{\mathbb{E}[(G_\beta(X) - G_\beta(\tilde{X}))^2]}{\mathbb{E}[(\Delta_\beta(X) - \Delta_\beta(\tilde{X}))^2]} \leqslant \frac{\mathbb{E}[(G_\beta(X) - G_\beta(\tilde{X}))^2]}{\mathbb{E}[(f_\beta \circ \Delta_\beta(X) - f_\beta \circ \Delta_\beta(\tilde{X}))^2]}.$$

Switching back to the notation $1 - \langle \hat{A}, \Theta \rangle$ and $1 - \langle \hat{A}, \tilde{\Theta} \rangle$ instead of $X$ and $\tilde{X}$ and letting $p(\theta)$ denote the probability density of $\Theta$, we note that the above ratio can be equivalently written as:

$$\frac{\iiint_{\mathcal{O}^3} \mathbb{E}[(G_\beta(1 - \langle \hat{A}, \Theta \rangle) - G_\beta(1 - \langle \hat{A}, \tilde{\Theta} \rangle))^2 \mid \hat{\Theta}, \Theta, \tilde{\Theta} \in \{\theta_0, \theta_1, \theta_2\}] \Pi_{i=1}^3 p(\theta_i) \, d\theta_0 \, d\theta_1 \, d\theta_2}{\iiint_{\mathcal{O}^3} \mathbb{E}[(f_\beta \circ \Delta_\beta(1 - \langle \hat{A}, \Theta \rangle) - f_\beta \circ \Delta_\beta(1 - \langle \hat{A}, \tilde{\Theta} \rangle))^2 \mid \hat{\Theta}, \Theta, \tilde{\Theta} \in \{\theta_0, \theta_1, \theta_2\}] \Pi_{i=1}^3 p(\theta_i) \, d\theta_0 \, d\theta_1 \, d\theta_2}.$$

In the next part, we will bound the ratio between $\mathbb{E}[(G_\beta(1 - \langle \hat{A}, \Theta \rangle) - G_\beta(1 - \langle \hat{A}, \tilde{\Theta} \rangle))^2]$ and $\mathbb{E}[(f_\beta \circ \Delta_\beta(1 - \langle \hat{A}, \Theta \rangle) - f_\beta \circ \Delta_\beta(1 - \langle \hat{A}, \tilde{\Theta} \rangle))^2]$, conditioned on $\hat{\Theta}, \Theta, \tilde{\Theta} \in \{\theta_0, \theta_1, \theta_2\}$, for any triplet $\theta_0, \theta_1, \theta_2 \in \mathcal{O}$, providing an upper bound on the above ratio. We denote this ratio $R(\beta)$. Without loss of generality, we consider $\theta_0, \theta_1, \theta_2 \in \mathcal{O}$ with associated conditional probability $p_0, p_1, p_2 \in ]0, 1]$, where $p_0 + p_1 + p_2 = 1$, and we let $a = 1 - \langle \theta_0, \theta_1 \rangle, b = 1 - \langle \theta_0, \theta_2 \rangle, c = 1 - \langle \theta_1, \theta_2 \rangle$. We introduce the notation $A = 2p_0 p_1 (p_0 + p_1)$, $B = 2p_0 p_2 (p_0 + p_2)$, $C = 2p_1 p_2 (p_1 + p_2)$, and $D = 2p_0 p_1 p_2$. To upper bound the ratio of expectation conditioned on $\hat{\Theta}, \Theta, \tilde{\Theta} \in \{\theta_0, \theta_1, \theta_2\}$, we have four cases to consider:

(i) $1 - \langle \theta_0, \theta_1 \rangle \leqslant \delta_\beta, 1 - \langle \theta_0, \theta_2 \rangle \leqslant \delta_\beta, 1 - \langle \theta_1, \theta_2 \rangle \leqslant \delta_\beta,$

(ii) $1 - \langle \theta_0, \theta_1 \rangle \geqslant \delta_\beta, 1 - \langle \theta_0, \theta_2 \rangle \leqslant \delta_\beta, 1 - \langle \theta_1, \theta_2 \rangle \leqslant \delta_\beta,$

(iii) $1 - \langle \theta_0, \theta_1 \rangle \geqslant \delta_\beta, 1 - \langle \theta_0, \theta_2 \rangle \geqslant \delta_\beta, 1 - \langle \theta_1, \theta_2 \rangle \leqslant \delta_\beta,$

(iv) $1 - \langle \theta_0, \theta_1 \rangle \geqslant \delta_\beta, 1 - \langle \theta_0, \theta_2 \rangle \geqslant \delta_\beta, 1 - \langle \theta_1, \theta_2 \rangle \geqslant \delta_\beta.$

In the case (i), we have $G_\beta(1 - \langle \theta_m, \theta_n \rangle) = f_\beta \circ \Delta_\beta(1 - \langle \theta_m, \theta_n \rangle)$ for $m, n \in \{1, 2, 3\}$ and the expectation over $G_\beta$ and $f_\beta \circ \Delta_\beta$ are equal. In the case (ii), we can write the ratio of conditional expectations as

$$\frac{(A + 2D)(\Delta_\beta(\delta_\beta)/\delta_\beta)^2 (a - \delta_\beta)^2}{A\Delta_\beta(\delta_\beta)^2 + B\Delta_\beta(b)^2 + C\Delta_\beta(c)^2 + D(\Delta_\beta(\delta_\beta) - \Delta_\beta(b))^2 + D(\Delta_\beta(\delta_\beta) - \Delta_\beta(c))^2 + D(\Delta_\beta(b) - \Delta_\beta(c))^2}.$$

We note that the numerator $N(a)$ is maximized for $a = 2$, where it takes the value $N(2) = \frac{\Delta_\beta(\delta_\beta)^2}{\delta_\beta^2}(A + 2D)(2 - \delta)^2$ and that the denominator $D(b, c)$ is minimized either by $b = c = 0$ if $D \leqslant B$ or $D \leqslant C$, with $D(0, 0) = \Delta_\beta(\delta_\beta)^2 (A + 2D)$. For $b = c = \delta_\beta$ to be a minimizer of $D(b, c)$ we would need to have simultaneously $B < D$ and $C < D$, implying that $p_1 > 1/2$ and $p_0 > 1/2$, which is not possible as $p_0 + p_1 < 1$. So the maximal ratio is $N(2)/D(0, 0)$ which is equal to $\frac{(2 - \delta_\beta)^2}{\delta_\beta^2}$.

In the case (iii), we can write the ratio of conditional expectations as:

$$\frac{(\Delta_\beta(\delta_\beta)/\delta_\beta)^2 ((A + D)(a - \delta_\beta)^2 + (B + D)(b - \delta_\beta)^2 + D(a - b)^2)}{A\Delta_\beta(\delta_\beta)^2 + B\Delta_\beta(\delta_\beta)^2 + C\Delta_\beta(c)^2 + 2D(\Delta_\beta(\delta_\beta) - \Delta_\beta(c))^2}.$$

We can verify that the numerator $N(a, b)$ is strictly convex and quadratic in $a, b$ as its Hessian is positive-definite. We can compute the Hessian by taking second partial derivatives:

$$\frac{\partial^2 N(a, b)}{\partial a^2} = 2A + 2D, \quad \frac{\partial^2 N(a, b)}{\partial b^2} = 2B + 2D, \quad \frac{\partial^2 N(a, b)}{\partial a\, \partial b} = \frac{\partial^2 N(a, b)}{\partial b\, \partial a} = -2D.$$

We then obtain the Hessian as $H_N = \begin{pmatrix} 2A + 2D & -2D \\ -2D & 2B + 2D \end{pmatrix}$ which is symmetric, and strictly diagonally dominant since both diagonal entries are strictly greater than the sum of the absolute values of the off-diagonal terms:

$$2A + 2D > 2D, \qquad 2B + 2D > 2D,$$

which holds as $A, B > 0$. Therefore, by Horn & Johnson (2012, Theorem 6.1.10), the Hessian is positive-definite, and thus $N(a, b)$ is strictly convex. Since $N(a, b)$ is strictly convex on the compact convex set $[\delta_\beta, 2] \times [\delta_\beta, 2]$, it attains its maximum at one of the extreme points of the domain. These extreme points are

$$(a, b) \in \{ (2, 2), (2, \delta_\beta), (\delta_\beta, 2), (\delta_\beta, \delta_\beta) \}.$$

We have that $N(\delta_\beta, \delta_\beta) = 0$ and for the other corners, we evaluate

$$N(2, \delta_\beta) = (A + 2D)\frac{\Delta_\beta(\delta_\beta)^2}{\delta_\beta^2}(2 - \delta_\beta)^2, \quad N(\delta_\beta, 2) = (B + 2D)\frac{\Delta_\beta(\delta_\beta)^2}{\delta_\beta^2}(2 - \delta_\beta)^2,$$

$$N(2, 2) = (A + B + 2D)\frac{\Delta_\beta(\delta_\beta)^2}{\delta_\beta^2}(2 - \delta_\beta)^2,$$

thus the numerator $N(a, b)$ is smaller than $N(2, 2) = \frac{\Delta_\beta(\delta_\beta)^2}{\delta_\beta^2}(2 - \delta_\beta)^2(A + B + 2D)$. Looking at the denominator $D(c) = (A+B)\Delta_\beta(\delta_\beta)^2 + C\Delta_\beta(c)^2 + 2D(\Delta_\beta(\delta_\beta) - \Delta_\beta(c))^2$, we note that it is minimized by $c = 0$ if $C \geqslant 2D$ and takes the value $D(0) = (A+B+2D)\Delta_\beta(\delta_\beta)^2$ and the ratio $N(2, 2)/D(0) = \frac{(2-\delta_\beta)^2}{\delta_\beta^2}$.

Lastly, we need to consider the case (iv), where the ratio of conditional expectations can be written as

$$\frac{A(a - \delta_\beta)^2 + B(b - \delta_\beta)^2 + C(c - \delta_\beta)^2 D(a - b)^2 + D(a - c)^2 + D(b - c)^2}{\delta_\beta^2(A + B + C)}.$$

We define $N(a, b, c)$ as the numerator, and compute the Hessian by taking second partial derivatives:

$$\frac{\partial^2 N}{\partial a^2} = 2A + 4D, \quad \frac{\partial^2 N}{\partial b^2} = 2B + 4D, \quad \frac{\partial^2 N}{\partial c^2} = 2C + 4D, \quad \frac{\partial^2 N}{\partial a \partial b} = \frac{\partial^2 N}{\partial a \partial c} = \frac{\partial^2 N}{\partial b \partial c} = -2D.$$

The resulting Hessian is:

$$H_N = \begin{pmatrix} 2A + 4D & -2D & -2D \\ -2D & 2B + 4D & -2D \\ -2D & -2D & 2C + 4D \end{pmatrix}.$$

This matrix is symmetric and strictly diagonally dominant since all diagonal entries are strictly greater than the sum of the absolute values of the off-diagonal entries:

$$2A + 4D > 4D, \quad 2B + 4D > 4D, \quad 2C + 4D > 4D,$$

which holds since $A, B, C > 0$. Therefore the Hessian is positive-definite, and thus $N(a, b, c)$ is strictly convex. The domain $a, b, c \in [\delta_\beta, 2]$ is restricted as the sum of the angles between the vectors $\theta_0, \theta_1, \theta_2$ is at most $2\pi$ (the maximum is achieved when $\theta_0, \theta_1, \theta_2$ are coplanar). We can express this condition as $\arccos(1 - a) + \arccos(1 - b) + \arccos(1 - c) \leqslant 2\pi$ with $1 - a, 1 - b, 1 - c \in [\delta_\beta - 1, 1]$. The function $\arccos(x)$ is concave for $x \in [0, 1]$, and the sum of concave functions is concave, so $\arccos(1-a) + \arccos(1-b) + \arccos(1-c)$

is concave. Then the superlevel set $\arccos(1-a) + \arccos(1-b) + \arccos(1-c) \leqslant 2\pi$ is convex, see Boyd & Vandenberghe (2004, Section 3.1.6).

As the function $N(a,b,c)$ is convex and the domain $a, b, c \in [\delta_\beta, 2]$ subject to $\arccos(1-a) + \arccos(1-b) + \arccos(1-c) \leqslant 2\pi$ is a convex set, it is maximized at points on the border of the domain. At the point $a = b = c = \delta_\beta$, we have $N(\delta_\beta, \delta_\beta, \delta_\beta) = 0$. As $a, b, c \geqslant \delta_\beta > 1$ (see Appendix D), it implies that $\arccos(1-a), \arccos(1-b), \arccos(1-c) \in ]\pi/2, \pi]$, thus the condition $\arccos(1-a) + \arccos(1-b) + \arccos(1-c) \leqslant 2\pi$ implies that no point $a, b, c$ can reach 2. The remaining extreme point is therefore when two points are equal to $\delta_\beta$ and the last one is such that $\arccos(1-a) + \arccos(1-b) + \arccos(1-c) = 2\pi$. We can assume without loss of generality that $A \geqslant B$ and $A \geqslant C$ and to simplify the exposition, we let $\theta_\beta = \arccos(1-\delta_\beta)$. The maximum of $N(a,b,c)$ is given by $N(t, \delta_\beta, \delta_\beta)$ where $t$ is such that $\arccos(1-t) = 2\pi - 2\theta_\beta$. We have that $t = 1 - \cos(2\pi - 2\theta_\beta) = 1 - \cos(2\theta_\beta) = 1 - (2\cos(\theta_\beta)^2 - 1) = 2 - 2(1-\delta_\beta)^2 = 2\delta_\beta(2-\delta_\beta)$. Thus we have that $N(t, \delta_\beta, \delta_\beta) = (A + 2D)(t - \delta_\beta)^2 = (A + 2D)(2\delta_\beta(2-\delta_\beta) - \delta_\beta)^2 = (A + 2D)\delta_\beta^2(3 - 2\delta_\beta)^2 \leqslant (A + 2D)(2 - \delta_\beta)^2$. We then use the fact that $2D \leqslant B + C$ for all $p_0, p_1, p_2 \in [0,1]$ to conclude that

$$\frac{N(a,b,c)}{\delta_\beta^2(A + B + C)} \leqslant \frac{(A + 2D)\delta_\beta^2(2 - \delta_\beta)^2}{\delta_\beta^2(A + B + C)} \leqslant \frac{(2 - \delta_\beta)^2}{\delta_\beta^2}.$$

From the analysis of (i), (ii), (iii), and (iv), we conclude that, for any $\theta_0, \theta_1, \theta_2 \in \mathcal{O}$, we have

$$\frac{\mathbb{E}\big[\big(G_\beta(1 - \langle \hat{A}, \Theta \rangle) - G_\beta(1 - \langle \hat{A}, \tilde{\Theta} \rangle)\big)^2 \mid \hat{\Theta}, \Theta, \tilde{\Theta} \in \{\theta_0, \theta_1, \theta_2\}\big]}{\mathbb{E}\big[\big(f_\beta \circ \Delta_\beta(1 - \langle \hat{A}, \Theta \rangle) - f_\beta \circ \Delta_\beta(1 - \langle \hat{A}, \tilde{\Theta} \rangle)\big)^2 \mid \hat{\Theta}, \Theta, \tilde{\Theta} \in \{\theta_0, \theta_1, \theta_2\}\big]} \leqslant \frac{(2 - \delta_\beta)^2}{\delta_\beta^2},$$

and therefore, it holds that

$$\frac{\mathbb{E}\big[\big(G_\beta(1 - \langle \hat{A}, \Theta \rangle) - G_\beta(1 - \langle \hat{A}, \tilde{\Theta} \rangle)\big)^2\big]}{\mathbb{E}\big[\big(f_\beta \circ \Delta_\beta(1 - \langle \hat{A}, \Theta \rangle) - f_\beta \circ \Delta_\beta(1 - \langle \hat{A}, \tilde{\Theta} \rangle)\big)^2\big]} \leqslant \frac{(2 - \delta_\beta)^2}{\delta_\beta^2}.$$

Using the above result in Equation (4), we get that

$$\frac{\mathbb{E}\left[\mathbb{V}\left[\bar{\Delta}_\beta\left(1 - \langle \hat{A}, \Theta \rangle\right) \mid \hat{\Theta}\right]\right]}{\mathbb{E}\left[\mathbb{V}\left[\Delta_\beta\left(1 - \langle \hat{A}, \Theta \rangle\right) \mid \hat{\Theta}\right]\right]} \leqslant \left(1 + \frac{2 - \delta_\beta}{\delta_\beta}\right)^2 = \frac{4}{\delta_\beta^2} \leqslant 4.$$

where we used that $\delta_\beta \in [1, 3/2]$ (see Appendix D for a detailed proof).

## C  Extension to general spaces

To extend the proof technique of Section 5.2 and Section 5.3, we first need to introduce the *alignment function* $\alpha(\hat{\theta}) := \max_{\theta \in \mathcal{O}} \langle \pi_\star(\hat{\theta}), \theta \rangle$ as well as *optimal environment* mapping $\rho(\hat{\theta}) = \arg\max_{\theta \in \mathcal{O}} \langle \pi_\star(\hat{\theta}), \theta \rangle$. We can define the *extended regret function* $\Delta_\beta(x, \alpha) := \phi_\beta(\alpha) - \phi_\beta(\alpha - x)$ and note that

$$\Delta_\beta(\alpha(\hat{\Theta}) - \langle \hat{A}, \Theta \rangle, \alpha(\hat{\Theta})) = \phi_\beta(\alpha(\hat{\Theta})) - \phi_\beta(\langle \hat{A}, \Theta \rangle).$$

Integrating over the Bernoulli outcome, we can write the expected regret using $\Delta_\beta(x, \alpha(\hat{\theta}))$:

$$\mathbb{E}[\Delta_\beta(\alpha(\hat{\Theta}) - \langle \hat{A}, \Theta \rangle, \alpha(\hat{\Theta}))] = \mathbb{E}[\phi_\beta(\langle \hat{A}, \rho(\hat{\Theta}) \rangle) - \phi_\beta(\langle \hat{A}, \Theta \rangle)] \geqslant \mathbb{E}[\phi_\beta(\langle A^\star, \Theta \rangle) - \phi_\beta(\langle \hat{A}, \Theta \rangle)],$$

where we used that $\hat{\Theta}$ and $\Theta$ are identically distributed and that the inequality $\langle \pi_\star(\theta), \rho(\theta) \rangle \geqslant \langle \pi_\star(\theta), \theta \rangle$ holds point-wise for all $\theta \in \mathcal{O}$ and therefore holds in expectation. Similarly to the proof in Section 5.2, we construct a function $\bar{\Delta}_\beta(x, \alpha(\hat{\theta}))$ as the tightest upper bound on $\Delta_\beta(x, \alpha(\hat{\theta}))$ that satisfies the requirements of Lemma 15.

**Definition 25 (Extended regret surrogate)** *For every $\hat{\theta} \in \mathcal{O}$, we construct the* extended regret surrogate *function $\bar{\Delta}_\beta(x, \alpha(\hat{\theta}))$ as the tightest upper bound on $\Delta_\beta(x, \alpha(\hat{\theta}))$ such that $\bar{\Delta}_\beta(x, \alpha(\hat{\theta}))/x$ is non-decreasing over $x \geqslant 0$.*

*Namely, let $\delta_\beta(\alpha(\hat{\theta})) = \arg\max_{x \in [0, 1+\alpha]} \frac{\Delta_\beta(x, \alpha(\hat{\theta}))}{x}$, we define the function $\bar{\Delta}_\beta(x, \alpha(\hat{\theta}))$ as*

$$
\bar{\Delta}_\beta(x, \alpha(\hat{\theta})) = \begin{cases} \Delta_\beta(x, \alpha(\hat{\theta})) & x \in [0, \delta_\beta(\alpha(\hat{\theta}))] \\ \Delta_\beta(\delta_\beta(\alpha(\hat{\theta})), \alpha(\hat{\theta})) + (x - \delta_\beta(\alpha(\hat{\theta}))) \cdot \kappa_\beta(\alpha(\hat{\theta})) & x \in ]\delta_\beta(\alpha(\hat{\theta})), 1 + \alpha(\hat{\theta})] \end{cases},
$$

*where $\kappa_\beta(\alpha(\hat{\theta})) = \Delta_\beta(\delta_\beta(\alpha(\hat{\theta})), \alpha(\hat{\theta}))/\delta_\beta(\alpha(\hat{\theta})) = \max_{x \in [0, 1+\alpha(\hat{\theta})]} \frac{\Delta_\beta(x, \alpha(\hat{\theta}))}{x}$.*

We are now equipped to extend Lemma 17 to general action and parameter spaces.

**Lemma 26** *Let the extended regret surrogate be defined as in Definition 25. Then, it holds that*

$$
\mathbb{E}[\mathrm{Bern}(\phi_\beta(\langle A^\star, \Theta \rangle)) - \mathrm{Bern}(\phi_\beta(\langle \hat{A}, \Theta \rangle))]^2 \leqslant 2d \cdot \mathbb{E}\left[ \mathbb{V}\left[ \bar{\Delta}_\beta(\alpha(\hat{\Theta}) - \langle \hat{A}, \Theta \rangle, \alpha(\hat{\Theta})) \mid \hat{\Theta} \right] \right].
$$

**Proof 27** *The proof follows closely the technique used to prove Lemma 17. We note that conditioned on $\hat{\Theta} = \hat{\theta}$, the extended regret surrogate is a mapping from $[0, 1 + \alpha(\hat{\theta})]$ to $[0, 1]$, that $\Delta_\beta(0, \alpha(\hat{\theta})) = \phi_\beta(\alpha(\hat{\theta})) - \phi_\beta(\alpha(\hat{\theta})) = 0$ and fulfills the assumptions of Lemma 15.*

Noting that $\mathbb{V}\left[ \phi_\beta(\langle \hat{A}, \Theta \rangle) \mid \hat{\Theta} \right] = \mathbb{V}\left[ \Delta_\beta(\alpha(\hat{\Theta}) - \langle \hat{A}, \Theta \rangle, \alpha(\hat{\Theta})) \mid \hat{\Theta} \right]$ and using Lemma 12, we have that the information ratio $\Gamma$ can be bounded by

$$
\Gamma \leqslant d \cdot \frac{\mathbb{E}\left[ \mathbb{V}\left[ \bar{\Delta}_\beta \left( \alpha(\hat{\Theta}) - \langle \hat{A}, \Theta \rangle, \alpha(\hat{\Theta}) \right) \mid \hat{\Theta} \right] \right]}{\mathbb{E}\left[ \mathbb{V}\left[ \Delta_\beta \left( \alpha(\hat{\Theta}) - \langle \hat{A}, \Theta \rangle, \alpha(\hat{\Theta}) \right) \mid \hat{\Theta} \right] \right]}.
$$

Similarly to the analysis for the case $\mathcal{O} \subseteq \mathcal{A}$, we continue the proof by noting that the ratio of expected variance can be written as a ratio of expected squared difference,

$$
\frac{\mathbb{E}\left[ \mathbb{V}\left[ \bar{\Delta}_\beta(\alpha(\hat{\Theta}) - \langle \hat{A}, \Theta \rangle, \alpha(\hat{\Theta})) \mid \hat{\Theta} \right] \right]}{\mathbb{E}\left[ \mathbb{V}\left[ \Delta_\beta(\alpha(\hat{\Theta}) - \langle \hat{A}, \Theta \rangle, \alpha(\hat{\Theta})) \mid \hat{\Theta} \right] \right]} = \frac{\mathbb{E}\left[ \mathbb{E}\left[ \left( \bar{\Delta}_\beta(\alpha(\hat{\Theta}) - \langle \hat{A}, \Theta \rangle, \alpha(\hat{\Theta})) - \bar{\Delta}_\beta(\alpha(\hat{\Theta}) - \langle \hat{A}, \tilde{\Theta} \rangle, \alpha(\hat{\Theta})) \right)^2 \mid \hat{\Theta} \right] \right]}{\mathbb{E}\left[ \mathbb{E}\left[ \left( \Delta_\beta(\alpha(\hat{\Theta}) - \langle \hat{A}, \Theta \rangle, \alpha(\hat{\Theta})) - \Delta_\beta(\alpha(\hat{\Theta}) - \langle \hat{A}, \tilde{\Theta} \rangle, \alpha(\hat{\Theta})) \right)^2 \mid \hat{\Theta} \right] \right]}
$$

$$
= \frac{\mathbb{E}\left[ \mathbb{E}\left[ \left( \bar{\Delta}_\beta(X, \alpha(\hat{\Theta})) - \bar{\Delta}_\beta(\tilde{X}, \alpha(\hat{\Theta})) \right)^2 \mid \hat{\Theta} \right] \right]}{\mathbb{E}\left[ \mathbb{E}\left[ \left( \Delta_\beta(X, \alpha(\hat{\Theta})) - \Delta_\beta(\tilde{X}, \alpha(\hat{\Theta})) \right)^2 \mid \hat{\Theta} \right] \right]},
$$

where $\tilde{\Theta}$ is a random variable independent and identically distributed as $\Theta$, and in the last equality we used a more compact notation $X = \alpha(\hat{\Theta}) - \langle \hat{A}, \Theta \rangle$ and $\tilde{X} = \alpha(\hat{\Theta}) - \langle \hat{A}, \tilde{\Theta} \rangle$.

Conditioned on $\hat{\Theta} = \hat{\theta}$, for $x \in [0, \delta_\beta(\alpha(\hat{\theta}))[$, we have $\bar{\Delta}_\beta(x, \alpha(\hat{\theta})) = \Delta_\beta(x, \alpha(\hat{\theta}))$ and for $x \in [\delta_\beta(\alpha(\hat{\theta})), 2]$, we have that $\bar{\Delta}_\beta(x, \alpha(\hat{\theta})) \leqslant \Delta_\beta(x, \alpha(\hat{\theta})) + \kappa_\beta(\alpha(\hat{\theta}))(x - \delta_\beta(\alpha(\hat{\theta})))$ as $\Delta_\beta(\delta_\beta(\alpha(\hat{\theta})), \alpha(\hat{\theta})) \leqslant \Delta_\beta(x, \alpha(\hat{\theta}))$. Using this observation we can upper bound the difference $\bar{\Delta}_\beta(x, \alpha(\hat{\theta})) - \Delta_\beta(x, \alpha(\hat{\theta}))$ by a function $G_\beta(x, \alpha(\hat{\theta}))$ such that

$$
G_\beta(x, \alpha(\hat{\theta})) = \begin{cases} 0 & x \in [0, \delta_\beta(\alpha(\hat{\theta}))[ \\ (x - \delta_\beta(\alpha(\hat{\theta})))\kappa_\beta(\alpha(\hat{\theta})) & x \in [\delta_\beta(\alpha(\hat{\theta})), 2] \end{cases}.
$$

We note that for all $x, y \in [0, 2]$, we have that

$$
|\Delta_\beta(x, \alpha(\hat{\theta})) + G_\beta(x, \alpha(\hat{\theta})) - \Delta_\beta(y, \alpha(\hat{\theta})) - G_\beta(y, \alpha(\hat{\theta}))| \geqslant |\bar{\Delta}_\beta(x, \alpha(\hat{\theta})) - \bar{\Delta}_\beta(y, \alpha(\hat{\theta}))|.
$$

Indeed, for $x, y \in [0, \delta_\beta(\alpha(\hat{\theta}))]$, we have $|\Delta_\beta(x, \alpha(\hat{\theta})) + G_\beta(x, \alpha(\hat{\theta})) - \Delta_\beta(y, \alpha(\hat{\theta})) - G_\beta(y, \alpha(\hat{\theta}))|$ and $|\bar{\Delta}_\beta(x, \alpha(\hat{\theta})) - \bar{\Delta}_\beta(y, \alpha(\hat{\theta}))|$ are equal to zero. Then for $x, y \in (\delta_\beta(\alpha(\hat{\theta})), 2]$, we have that

$$|\Delta_\beta(x, \alpha(\hat{\theta})) + G_\beta(x, \alpha(\hat{\theta})) - \Delta_\beta(y, \alpha(\hat{\theta})) - G_\beta(y, \alpha(\hat{\theta}))| = |\Delta_\beta(x, \alpha(\hat{\theta})) - \Delta_\beta(y, \alpha(\hat{\theta})) + \kappa_\beta(\alpha(\hat{\theta}))(x - y)|$$
$$\geqslant |\bar{\Delta}_\beta(x, \alpha(\hat{\theta})) - \bar{\Delta}_\beta(y, \alpha(\hat{\theta}))|,$$

and for $x \in (\delta_\beta(\alpha(\hat{\theta})), 2]$, $y \in [0, \delta_\beta(\alpha(\hat{\theta}))]$, we have again

$$|\Delta_\beta(x, \alpha(\hat{\theta})) + G_\beta(x, \alpha(\hat{\theta})) - \Delta_\beta(y, \alpha(\hat{\theta})) - G_\beta(y, \alpha(\hat{\theta}))| = |\Delta_\beta(x, \alpha(\hat{\theta})) - \Delta_\beta(y, \alpha(\hat{\theta})) + \kappa_\beta(\alpha(\hat{\theta}))(x - \delta_\beta(\alpha(\hat{\theta})))|$$
$$\geqslant |\bar{\Delta}_\beta(x, \alpha(\hat{\theta})) - \bar{\Delta}_\beta(y, \alpha(\hat{\theta}))|.$$

As the inequality $(\Delta_\beta(x, \alpha(\hat{\theta})) + G_\beta(x, \alpha(\hat{\theta})) - \Delta_\beta(y, \alpha(\hat{\theta})) - G_\beta(y, \alpha(\hat{\theta})))^2$ is greater than or equal to $(\bar{\Delta}_\beta(x, \alpha(\hat{\theta})) - \bar{\Delta}_\beta(y, \alpha(\hat{\theta})))^2$ holds everywhere, it holds also in expectation, and we have that

$$\frac{\mathbb{E}\left[\left(\bar{\Delta}_\beta(X, \alpha(\hat{\Theta})) - \bar{\Delta}_\beta(\tilde{X}, \alpha(\hat{\Theta}))\right)^2\right]}{\mathbb{E}\left[\left(\Delta_\beta(X, \alpha(\hat{\Theta})) - \Delta_\beta(\tilde{X}, \alpha(\hat{\Theta}))\right)^2\right]} \leqslant \frac{\mathbb{E}\left[\left(\Delta_\beta(X, \alpha(\hat{\Theta})) - \Delta_\beta(\tilde{X}, \alpha(\hat{\Theta})) + G_\beta(X, \alpha(\hat{\Theta})) - G_\beta(\tilde{X}, \alpha(\hat{\Theta}))\right)^2\right]}{\mathbb{E}\left[\left(\Delta_\beta(X, \alpha(\hat{\Theta})) - \Delta_\beta(\tilde{X}, \alpha(\hat{\Theta}))\right)^2\right]}$$

$$= 1 + 2\frac{\mathbb{E}\left[\left(\Delta_\beta(X, \alpha(\hat{\Theta})) - \Delta_\beta(\tilde{X}, \alpha(\hat{\Theta}))\right)\left(G_\beta(X, \alpha(\hat{\Theta})) - G_\beta(\tilde{X}, \alpha(\hat{\Theta}))\right)\right]}{\mathbb{E}\left[\left(\Delta_\beta(X, \alpha(\hat{\Theta})) - \Delta_\beta(\tilde{X}, \alpha(\hat{\Theta}))\right)^2\right]} + \frac{\mathbb{E}\left[\left(G_\beta(X, \alpha(\hat{\Theta})) - G_\beta(\tilde{X}, \alpha(\hat{\Theta}))\right)^2\right]}{\mathbb{E}\left[\left(\Delta_\beta(X, \alpha(\hat{\Theta})) - \Delta_\beta(\tilde{X}, \alpha(\hat{\Theta}))\right)^2\right]}$$

$$\leqslant 1 + 2\sqrt{\frac{\mathbb{E}\left[\left(G_\beta(X, \alpha(\hat{\Theta})) - G_\beta(\tilde{X}, \alpha(\hat{\Theta}))\right)^2\right]}{\mathbb{E}\left[\left(\Delta_\beta(X, \alpha(\hat{\Theta})) - \Delta_\beta(\tilde{X}, \alpha(\hat{\Theta}))\right)^2\right]}} + \frac{\mathbb{E}\left[\left(G_\beta(X, \alpha(\hat{\Theta})) - G_\beta(\tilde{X}, \alpha(\hat{\Theta}))\right)^2\right]}{\mathbb{E}\left[\left(\Delta_\beta(X, \alpha(\hat{\Theta})) - \Delta_\beta(\tilde{X}, \alpha(\hat{\Theta}))\right)^2\right]}, \quad (5)$$

where in the second inequality, we used Cauchy-Schwarz inequality. We can then focus on studying the ratio of expected squared differences between $G_\beta$ and $\Delta_\beta$. We will first apply a transformation $f(x) = \min(\Delta_\beta(\delta_\beta(\alpha(\hat{\theta})), \alpha(\hat{\theta})), x)$ to crop the values of $\Delta_\beta$ that are above $\Delta_\beta(\delta_\beta(\alpha(\hat{\theta})), \alpha(\hat{\theta}))$. We can write the resulting function $f \circ \Delta_\beta$ as

$$f \circ \Delta_\beta(x, \alpha(\hat{\theta})) = \begin{cases} \Delta_\beta(x, \alpha(\hat{\theta})) & x \in [0, \delta_\beta(\alpha(\hat{\theta}))[ \\ \Delta_\beta(\delta_\beta(\alpha(\hat{\theta})), \alpha(\hat{\theta})) & x \in [\delta_\beta(\alpha(\hat{\theta})), 2[ \end{cases}$$

We observe that the transformation $f$ contracts the function $\Delta_\beta$ as, for all $x, y \in [0, 2]$, we have $|f \circ \Delta_\beta(x, \alpha(\hat{\theta})) - f \circ \Delta_\beta(y, \alpha(\hat{\theta}))| \leqslant |\Delta_\beta(x, \alpha(\hat{\theta})) - \Delta_\beta(y, \alpha(\hat{\theta}))|$. As the inequality holds everywhere, it holds also in expectation and we have that $\mathbb{E}\left[\left(f \circ \Delta_\beta(X, \alpha(\hat{\Theta})) - f \circ \Delta_\beta(\tilde{X}, \alpha(\hat{\Theta}))\right)^2\right] \leqslant \mathbb{E}\left[\left(\Delta_\beta(X, \alpha(\hat{\Theta})) - \Delta_\beta(\tilde{X}, \alpha(\hat{\Theta}))\right)^2\right]$ and it follows that

$$\frac{\mathbb{E}\left[\left(G_\beta(X, \alpha(\hat{\Theta})) - G_\beta(\tilde{X}, \alpha(\hat{\Theta}))\right)^2\right]}{\mathbb{E}\left[\left(\Delta_\beta(X, \alpha(\hat{\Theta})) - \Delta_\beta(\tilde{X}, \alpha(\hat{\Theta}))\right)^2\right]} \leqslant \frac{\mathbb{E}\left[\left(G_\beta(X, \alpha(\hat{\Theta})) - G_\beta(\tilde{X}, \alpha(\hat{\Theta}))\right)^2\right]}{\mathbb{E}\left[\left(f \circ \Delta_\beta(X, \alpha(\hat{\Theta})) - f \circ \Delta_\beta(\tilde{X}, \alpha(\hat{\Theta}))\right)^2\right]}.$$

Similarly to Lemma 19, we will bound the ratio conditioned on $\hat{\Theta}, \Theta, \tilde{\Theta} \in \{\theta_0, \theta_1, \theta_2\}$, for any triplet $\theta_0, \theta_1, \theta_2 \in \mathcal{O}$, in order to provide an upper bound on the ratio when taking the expectations over $\theta_0, \theta_1, \theta_2 \in \mathcal{O}$.

Using a very similar proof to the one for Lemma 19, we can show that

$$\frac{\mathbb{E}\left[\left(G_\beta(X, \alpha(\hat{\Theta})) - G_\beta(\tilde{X}, \alpha(\hat{\Theta}))\right)^2 \mid \hat{\Theta}, \Theta, \tilde{\Theta} \in \{\theta_0, \theta_1, \theta_2\}\right]}{\mathbb{E}\left[\left(f \circ \Delta_\beta(X, \alpha(\hat{\Theta})) - f \circ \Delta_\beta(\tilde{X}, \alpha(\hat{\Theta}))\right)^2 \mid \hat{\Theta}, \Theta, \tilde{\Theta} \in \{\theta_0, \theta_1, \theta_2\}\right]} \leqslant \frac{4\kappa_\beta(\alpha)^2(1 + \alpha - \delta_\beta(\alpha))^2}{\Delta_\beta(\delta_\beta(\alpha), \alpha)^2},$$

where $\alpha = \min_{\hat{\theta} \in \mathcal{O}} \alpha(\hat{\theta}) \leqslant 1$. Then using the definitions of $\kappa_\beta(\alpha)$ and $\Delta_\beta(\alpha, \alpha)$, we can simplify the right-hand side of the above inequality as

$$\frac{4\kappa_\beta(\alpha)^2(1 + \alpha - \delta_\beta(\alpha))^2}{\Delta_\beta(\delta_\beta(\alpha), \alpha)^2} = \frac{4(1 + \alpha - \delta_\beta(\alpha))^2}{\delta_\beta(\alpha)^2} \leqslant \frac{(4 - 2\delta_\beta(\alpha))^2}{\delta_\beta(\alpha)^2}.$$

Plugging the above result in Equation (5), we get that

$$\frac{\mathbb{E}\left[\mathbb{V}\left[\bar{\Delta}_\beta\left(\alpha(\hat{\Theta}) - \langle \hat{A}, \Theta \rangle, \hat{\Theta}\right) \mid \hat{\Theta}\right]\right]}{\mathbb{E}\left[\mathbb{V}\left[\Delta_\beta\left(\alpha(\hat{\Theta}) - \langle \hat{A}, \Theta \rangle, \hat{\Theta}\right) \mid \hat{\Theta}\right]\right]} \leqslant \left(1 + \frac{4 - 2\delta_\beta(\alpha)}{\delta_\beta(\alpha)}\right)^2 \leqslant \frac{4^2}{\delta_\beta(\alpha)^2} \leqslant \frac{4^2}{\alpha^2},$$

where we used the fact that $\delta_\beta(\alpha) \geqslant \alpha$ (see Remark 28). This result concludes the proof of Proposition 4.

**Remark 28** *Remark about $a \leqslant \delta_\beta(a)$ for all $a \in [0,1]$.*

*We recall that $\Delta_\beta(x,a) = \phi_\beta(a) - \phi_\beta(a - x)$ and start by computing the derivative of $\Delta_\beta(x,a)/x$:*

$$\frac{d}{dx}\left(\frac{\Delta_\beta(x,a)}{x}\right) = \frac{\phi'_\beta(a-x)x - (\phi_\beta(a) - \phi_\beta(a-x))}{x^2}.$$

*Let $N(x,a)$ be the numerator of the above ratio, the first-order condition for the maximizer can therefore be written as*

$$N(x,a) = \phi_\beta(a) - \phi_\beta(a-x) - x\,\phi'_\beta(a-x),$$

*and the equality $N(x,a) = 0$ is achieved for $x = \delta_\beta(a)$. We observe that $N(0,a) = \phi_\beta(a) - \phi_\beta(a) = 0$. We will now show that $N(x,a)$ is strictly decreasing for $x \in [0,a]$ (thus $N(x,a) < 0$ for $x \in ]0,a]$) and strictly increasing for $x \geqslant a$, which implies that the equation $N(x,a) = 0$ has a unique positive solution and that this solution is strictly larger than $a$.*

*To do this, we study the sign of $d/dx N(x,a) = x\,\phi''_\beta(a-x)$ for $x \in [0,a]$ and for $x \geqslant a$. As $x \geqslant 0$ is always non-negative, the sign of the derivative is given by the sign of $\phi''_\beta(a-x)$. We recall that $\phi''_\beta(z) = \beta^2 \phi_\beta(z)(1 - \phi_\beta(z))(1 - 2\phi_\beta(z))$. As $\phi_\beta(z) \in [0,1]$ for all $z \in \mathbb{R}$, we have that $\phi''_\beta(z)$ is positive if $\phi_\beta(z) < 1/2$ and negative if $\phi_\beta(z) > 1/2$. As $\phi_\beta(0) = 1/2$ and $\phi_\beta(z)$ is increasing, we have that $\phi_\beta(z) < 1/2$ for all $z < 0$ and $\phi_\beta(z) > 1/2$ for all $z > 0$. Therefore, for $x \in [0,a]$, we have that $a - x \geqslant 0$ and $\phi''_\beta(a-x) \leqslant 0$, which implies that $N(x,a)$ is strictly decreasing for $x \in [0,a]$. Similarly, for $x \geqslant a$, we have that $a - x \leqslant 0$ and $\phi''_\beta(a-x) \geqslant 0$, which implies that $N(x,a)$ is strictly increasing for $x \geqslant a$.*

# D   Analyzing the functions $\Delta_\beta(\delta_\beta)$ and $\delta_\beta$

In this section, we study the quantities $\Delta_\beta(\delta_\beta)/\delta_\beta$ and $\delta_\beta$, illustrated respectively on Figure 4 and Figure 5. We will prove several important properties of $\delta_\beta$, $\Delta_\beta(\delta_\beta)/\delta_\beta$, which are used in the proof of Lemma 19. First, we prove that the function $\kappa_\beta(x) := \Delta_\beta(x)/x$ has a unique maximizer $\delta_\beta$ which lies in between $]1,2]$. Then we prove that $\delta_\beta$ is a decreasing function of $\beta$. We then analyze its limits and show that $\lim_{\beta \to 0^+} \delta_\beta = 3/2$ and $\lim_{\beta \to \infty} \delta_\beta = 1$, thus refining the domain of $\delta_\beta$ to $[1, 3/2]$. After that we analyze the monotonicity of the function $\Delta_\beta(\delta_\beta)/\delta_\beta$ and show that it is an increasing function of $\beta$.

## D.1   Uniqueness of the maximizer $\delta_\beta$

To see rigorously that $\kappa_\beta(x)$ has exactly one maximum in $[0,2]$ and that this unique maximum $\delta_\beta$ lies in the interval $[1,2]$, we will follow a similar technique as in Remark 28, that is studying the derivative of $\kappa_\beta(x)$ to derive optimality conditions and showing that these conditions are satisfied by only one point in $[1,2]$. We start by computing the derivative of $\kappa_\beta(x)$:

$$\kappa'_\beta(x) = \left(\frac{\Delta_\beta(x)}{x}\right)' = \frac{\phi'_\beta(1-x)x - (\phi_\beta(1) - \phi_\beta(1-x))}{x^2},$$

where we used that $\Delta'_\beta(x) = \phi'_\beta(1-x)$ and $\Delta_\beta(x) = \phi_\beta(1) - \phi_\beta(1-x)$.

Let $N(x) = \phi'_\beta(1-x)x + \phi_\beta(1-x) - \phi_\beta(1)$ be the numerator of $\kappa'_\beta(x)$. Since the denominator $x^2$ is strictly positive for all $x > 0$, the sign of $\kappa'_\beta(x)$ is determined by the sign of $N(x)$ and the sign of $N(x)$ has to reverse

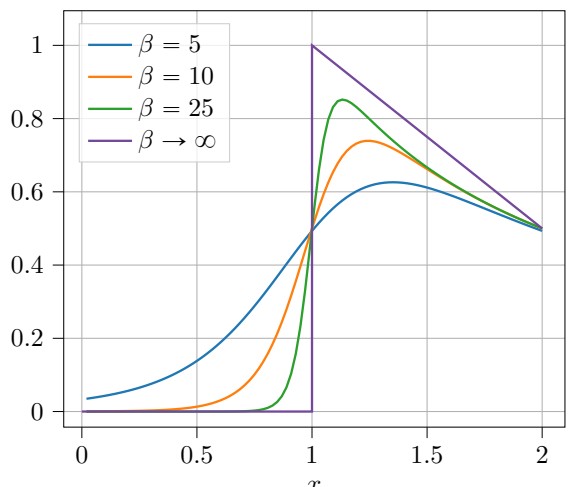 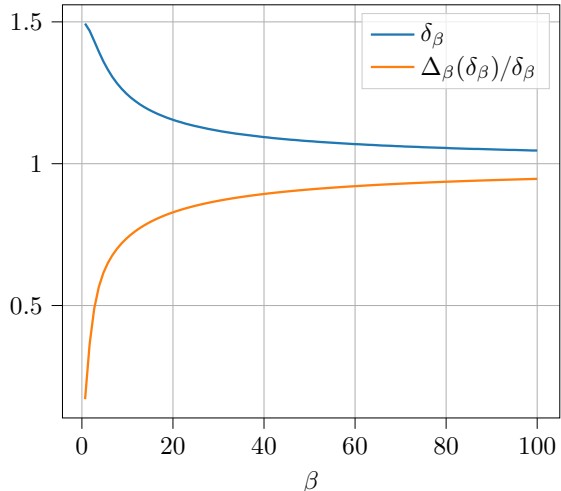

Figure 4: Illustration of $\kappa_\beta(x) = \Delta_\beta(x)/x$ over $x \in [0, 2]$ for different values of $\beta$. The maximum of the function is attained at $x = \delta_\beta$.

Figure 5: Illustration of $\delta_\beta$ and $\Delta_\beta(\delta_\beta)/\delta_\beta$ as functions of $\beta$. One can observe that $\delta_\beta$ decreases with $\beta$ while $\Delta_\beta(\delta_\beta)/\delta_\beta$ increases.

at the maximizer $\delta_\beta$. We will start by studying the derivative $N'(x) = -\phi_\beta''(1-x)x$. As $x \in ]0, 2]$ is positive, the sign of $N'(x)$ follows that of $-\phi_\beta''(1-x) = -\beta^2 \phi_\beta(1-x)(1-\phi_\beta(1-x))(1-2\phi_\beta(1-x))$, and we have that $N'(x)$ is positive when $\phi_\beta(1-x) < 1/2$, that is for $x < 1$ and negative when $\phi_\beta(1-x) > 1/2$, that is for $x > 1$. We therefore have that $N(x)$ is increasing over the interval $[0, 1[$, reaches a maximum at $x = 1$, and decreases over the interval $]1, 2]$. We now evaluate $N(x)$ at the points $x = 0$, $x = 1$, $x = 2$. We have that $N(0) = \phi_\beta(1) - \phi_\beta(1) = 0$ and $N(1) = \phi_\beta'(0) + \phi_\beta(0) - \phi_\beta(1) = \beta/4 + 1/2 - \phi_\beta(1)$. Now let $f(\beta) = \phi_\beta(1)$, we have that $f'(\beta) = f(\beta)(1 - f(\beta))$ is maximal for $\beta = 0$ and for $\beta > 0$, we have that $f(\beta) < f(0) + f'(0)\beta = 1/2 + \beta/4$, therefore we have that $N(1) > 0$ for $\beta > 0$. As $N(0) = 0$, $N(1) > 0$ and $N'(x) \geqslant 0$ for $x \in [0, 1]$, we have that $N(x)$ is positive over the interval $[0, 1]$. As $N'(x)$ is negative for $x \in ]1, 2]$, by continuity of the function, we only have to show that $N(2) < 0$ to prove that there exists a point $\delta_\beta$ in $[1, 2]$ at which $N(x)$ changes sign. We have that $N(2) = 2\phi_\beta'(-1) + \phi_\beta(-1) - \phi_\beta(1)$. Then using that $\phi_\beta(-1) = 1 - \phi_\beta(1)$ and that $\phi_\beta'(-1) = \beta\phi_\beta(1)(1 - \phi_\beta(1))$, we have that $N(2) = 2\beta\phi_\beta(1)(1 - \phi_\beta(1)) + 1 - 2\phi_\beta(1)$. As $\phi_\beta(1) \in [1/2, 1[$, we have that $\phi_\beta(1)(1 - \phi_\beta(1)) \leqslant 1/4$ and therefore $N(2) \leqslant \beta/2 + 1 - 2\phi_\beta(1) = 2(\beta/4 + 1/2 - \phi_\beta(1)) < 0$, using the same argument as previously. As $N(x)$ is positive over $[0, 1]$, strictly decreasing over $]1, 2]$, and negative at $x = 2$, there exists a unique point $\delta_\beta \in [1, 2]$ such that $N(\delta_\beta) = 0$. We therefore have that $\kappa_\beta(x)$ has a unique maximum $\delta_\beta$ in $[1, 2]$.

### D.2 On the monotonicity of $\delta_\beta$

We now study how the quantity $\delta_\beta$ varies with $\beta$. To this end, we analyze the total derivative of the first-order optimality condition satisfied by $\delta_\beta$, which takes the form $N(x, \beta) = \phi_\beta'(1-x)x + \phi_\beta(1-x) - \phi_\beta(1)$.

Since $\delta_\beta$ is defined by the condition $N(\delta_\beta, \beta) = 0$, we may differentiate this identity with respect to $\beta$ and apply the chain rule. We obtain

$$0 = \frac{d}{d\beta}N(\delta_\beta, \beta) = \partial_x N(\delta_\beta, \beta)\frac{d\delta_\beta}{d\beta} + \partial_\beta N(\delta_\beta, \beta) \quad \text{implying that} \quad \frac{d\delta_\beta}{d\beta} = -\frac{\partial_\beta N(\delta_\beta, \beta)}{\partial_x N(\delta_\beta, \beta)}.$$

We now compute the partial derivatives appearing in the above expression. Using the identity $\phi_\beta'(x) = \beta\phi_\beta(x)(1 - \phi_\beta(x))$ and the chain rule, we obtain $\partial_x N(x, \beta) = -x\phi_\beta''(1-x)$, where the second derivative of $\phi_\beta$ is given by $\phi_\beta''(x) = \beta^2\phi_\beta(x)(1 - \phi_\beta(x))(1 - 2\phi_\beta(x))$. Since $\delta_\beta > 1$, it follows that $\phi_\beta(1 - \delta_\beta) < 1/2$, and therefore $1 - 2\phi_\beta(1 - \delta_\beta) > 0$. This implies that $\phi_\beta''(1 - \delta_\beta) > 0$, and hence $\partial_x N(\delta_\beta, \beta) < 0$.

We now compute the partial derivative with respect to $\beta$ and we obtain

$$\partial_\beta N(x, \beta) = x\partial_\beta \phi'_\beta(1-x) + \partial_\beta \phi_\beta(1-x) - \partial_\beta \phi_\beta(1),$$

where $\partial_\beta \phi'_\beta(x) = \phi_\beta(x)(1-\phi_\beta(x))\left(1+\beta x(1-2\phi_\beta(x))\right)$ and $\partial_\beta \phi_\beta(x) = x\phi_\beta(x)(1-\phi_\beta(x))$. Substituting these expressions into the formula for $\partial_\beta N(x, \beta)$ and simplifying gives

$$\partial_\beta N(\delta_\beta, \beta) = \phi_\beta(1-\delta_\beta)(1-\phi_\beta(1-\delta_\beta))(1+\beta\,\delta_\beta(1-\delta_\beta)(1-2\phi_\beta(1-\delta_\beta))) - \phi_\beta(1)(1-\phi_\beta(1)).$$

We now let $q = \phi_\beta(1-\delta_\beta)$ and $p = \phi_\beta(1)$ and plug $x = \delta_\beta$ into the expression. Noting that $\partial_\beta \phi_\beta(x) = x\phi_\beta(x)(1-\phi_\beta(x))$, we obtain $\partial_\beta N(\delta_\beta, \beta) = q(1-q)(1+\beta\delta_\beta(1-\delta_\beta)(1-2q)) - p(1-p)$. We then recall that the first-order condition $N(\delta_\beta, \beta) = 0$ implies the identity $p - q = \beta\delta_\beta q(1-q)$, which we may use to eliminate $p$ and simplify the expression for $\partial_\beta N(\delta_\beta, \beta)$. A short algebraic manipulation gives

$$\partial_\beta N(\delta_\beta, \beta) = \beta\delta_\beta^2 q(1-q)\left(\beta q(1-q) - (1-2q)\right).$$

The first part of the expression $\beta\delta_\beta^2 q(1-q)$ is positive. We then have to analyze the sign of $\beta q(1-q)-(1-2q)$. Since $\delta_\beta > 1$, we have that $q < 1/2$, and it follows that $1 - 2q > 0$. Furthermore, from the identity $p - q = \beta\delta_\beta q(1-q)$ and the fact that $p < 1$, we have $\beta\delta_\beta q(1-q) < 1 - q \leqslant 1 - 2q$. As $\delta_\beta > 1$, the inequality $\beta q(1-q) < 1 - 2q$ follows. We conclude that $\beta q(1-q) - (1-2q)$ is negative, and so $\partial_\beta N(\delta_\beta, \beta) < 0$. Combined with the fact that $\partial_x N(\delta_\beta, \beta) < 0$, we obtain

$$\frac{d\delta_\beta}{d\beta} = -\frac{\partial_\beta N(\delta_\beta, \beta)}{\partial_x N(\delta_\beta, \beta)} < 0,$$

which shows that the value of $\delta_\beta$ is strictly decreasing with increasing values of $\beta$.

### D.3 Evaluating $\delta_\beta$ at the limits $\beta \to 0^+$ and $\beta \to \infty$

We now turn to the analysis of the limits of $\delta_\beta$ as $\beta \to 0^+$ and $\beta \to \infty$. We begin with the small $\beta$ regime. For small $\beta$, we use Taylor expansions around $\beta = 0$ and obtain

$$\phi_\beta(x) = \frac{1}{2} + \frac{\beta x}{4} - \frac{(\beta x)^3}{48} + O(\beta^5), \qquad \text{and} \qquad \phi'_\beta(x) = \frac{\beta}{4} - \frac{\beta^3 x^2}{16} + O(\beta^5).$$

We plug these expansions into the first-order condition $N(x, \beta) = 0$ and obtain,

$$N(x, \beta) = \frac{\beta^3}{48} x^2(3 - 2x) + O(\beta^5).$$

Dividing by $\beta^3$ and taking the limit as $\beta \to 0^+$ shows that $\delta_\beta$ must satisfy $x^2(3-2x) = 0$. The only solution to this equation in the interval $[1, 2]$ is $x = 3/2$, and so we conclude that $\lim_{\beta \to 0^+} \delta_\beta = \frac{3}{2}$. We now analyze the limit $\beta \to \infty$. Suppose by contradiction that there exists $\varepsilon > 0$ and a sequence $\beta_n \to \infty$ such that $\delta_{\beta_n} \geqslant 1 + \varepsilon$ for all $n$. Then, since $\phi_{\beta_n}(1-\delta_{\beta_n}) \leqslant \exp(-\beta_n\varepsilon)$, it follows that $\phi_{\beta_n}(1-\delta_{\beta_n}) \to 0$ as $\beta_n$ tends to $\infty$. But from the first-order condition

$$\phi_\beta(1) - \phi_\beta(1-\delta_\beta) = \beta\delta_\beta\phi_\beta(1-\delta_\beta)(1-\phi_\beta(1-\delta_\beta)),$$

the left-hand side tends to 1 while the right-hand side tends to 0, which is a contradiction. Hence, we must have

$$\lim_{\beta \to \infty} \delta_\beta = 1.$$

Combined with the monotonicity of $\delta_\beta$ in $\beta$, we refine its domain to $\delta_\beta \in [1, 3/2]$ for all $\beta > 0$. These results are in agreement with the numerical results illustrated in Figure 5.

# E   Adapting information ratio bounds from linear to logistic bandits

This section presents how to rigorously adapt the TS information bound from Dong et al. (2019, Proposition 17) to the *one-step compressed Thompson Sampling* such that it can be combined with the regret bound from Dong & Van Roy (2018, Theorem 1) or with Gouverneur et al. (2025, Theorem 1).

**Proposition 29 (Dong et al. (2019, Proposition 17) extended to *one-step compressed TS*)** *For all $\beta > 0$, under the logistic bandit setting with logistic function $\phi_\beta(x)$, letting $\tilde{\Theta}_t^\star$ and $\tilde{\Theta}_t$ satisfy the conditions in Dong & Van Roy (2018, Proposition 2), the* one-step compressed TS *information ratio is bounded as*

$$\tilde{\Gamma}(\tilde{\Theta}_t^\star, \tilde{\Theta}_t) := \frac{\mathbb{E}_t[R(\pi_\star(\tilde{\Theta}_t^\star), \Theta) - R(\pi_\star(\tilde{\Theta}_t), \Theta)]^2}{\mathrm{I}_t(\tilde{\Theta}_t^\star; R(\pi_\star(\tilde{\Theta}_t), \Theta), \tilde{\Theta}_t)} \leqslant d\frac{(1 + \exp(\beta)^4}{32\exp(\beta)^2}.$$

**Proof 30** *We combine the proof techniques from Dong et al. (2019, Proposition 17) and Dong & Van Roy (2018, Proposition 3). To simplify the exposition we will omit the subscript t and reuse the notation introduced in Section 5. With these notation, the one-step compressed information ratio can be written as*

$$\frac{\mathbb{E}[\mathrm{Bern}(\phi_\beta(\langle\pi_\star(\tilde{\Theta}^\star), \Theta\rangle)) - \mathrm{Bern}(\phi_\beta(\langle\pi_\star(\tilde{\Theta}), \Theta\rangle))]^2}{\mathrm{I}(\tilde{\Theta}^\star; \mathrm{Bern}(\phi_\beta(\langle\pi_\star(\tilde{\Theta}), \Theta\rangle)), \tilde{\Theta})}.$$

*Similarly to Dong et al. (2019, Proposition 17), we let $L_1 = \inf_{x\in[-1,1]} |\phi_\beta(x)'| = \frac{\beta\exp(\beta)}{(1+\exp(\beta))^2}$ and $L_2 = \sup_{x\in[-1,1]} |\phi_\beta(x)'| = \frac{\beta}{4}$. We will start by analyzing the numerator which we write as*

$$\mathbb{E}[\mathrm{Bern}(\phi_\beta(\langle\pi_\star(\tilde{\Theta}^\star), \Theta\rangle)) - \mathrm{Bern}(\phi_\beta(\langle\pi_\star(\tilde{\Theta}), \Theta\rangle))]^2 = \mathbb{E}[\phi_\beta(\langle\pi_\star(\tilde{\Theta}^\star), \Theta\rangle) - \phi_\beta(\langle\pi_\star(\tilde{\Theta}), \Theta\rangle)]^2$$
$$\leqslant L_2^2 \cdot \mathbb{E}[\langle\pi_\star(\tilde{\Theta}^\star), \Theta\rangle - \langle\pi_\star(\tilde{\Theta}), \Theta\rangle]^2,$$

*where the inequality follows from $\frac{\phi_\beta(x_1) - \phi_\beta(x_2)}{x_1 - x_2} \leqslant \sup_{x\in[-1,1]} |\phi_\beta(x)'|$ for all $x_1, x_2 \in [-1, 1]$.*

*We then focus on the denominator, which we lower bound similarly as in Dong & Van Roy (2018, Lemma 2). Introducing the notation $R(\tilde{\theta}) = \mathrm{Bern}(\phi_\beta(\langle\pi_\star(\tilde{\theta}), \Theta\rangle))$, we can write*

$$\mathrm{I}(\tilde{\Theta}^\star; \mathrm{Bern}(\phi_\beta(\langle\pi_\star(\tilde{\Theta}), \Theta\rangle)), \tilde{\Theta}) \overset{(i)}{=} \mathrm{I}(\tilde{\Theta}^\star; \tilde{\Theta}) + \mathrm{I}(\tilde{\Theta}^\star; \mathrm{Bern}(\phi_\beta(\langle\pi_\star(\tilde{\Theta}), \Theta\rangle))|\tilde{\Theta})$$
$$\overset{(j)}{=} \sum_{\tilde{\theta}\in\mathcal{O}_\epsilon} \sum_{\tilde{\theta}^\star\in\mathcal{O}_\epsilon} \mathbb{P}[\tilde{\Theta}=\tilde{\theta}]\mathbb{P}[\tilde{\Theta}^\star=\tilde{\theta}^\star]D_{\mathrm{KL}}\mathbb{P}_{R(\tilde{\theta})|\tilde{\Theta}^\star=\tilde{\theta}^\star}\mathbb{P}_{R(\tilde{\theta})}$$
$$\overset{(k)}{\geqslant} \sum_{\tilde{\theta}\in\mathcal{O}_\epsilon} \sum_{\tilde{\theta}^\star\in\mathcal{O}_\epsilon} \mathbb{P}[\tilde{\Theta}=\tilde{\theta}]\mathbb{P}[\tilde{\Theta}^\star=\tilde{\theta}^\star]2\big(\mathbb{E}[R(\tilde{\theta})|\tilde{\Theta}^\star=\tilde{\theta}^\star] - \mathbb{E}[R(\tilde{\theta})]\big)^2,$$

*where $\mathcal{O}_\epsilon$ is the set of values for the random variables $\tilde{\theta}^\star$ and $\tilde{\Theta}_t$ as defined in Dong & Van Roy (2018, Proposition 2). The inequality (i) follows from the chain-rule; (j) follows from $\tilde{\Theta}_t^\star$ and $\tilde{\Theta}_t$ being independent as they satisfy the conditions in Dong & Van Roy (2018, Proposition 2); (k) is obtained using the Donsker-Varadhan inequality (Gray, 2013, Theorem 5.2.1) as in Russo & Van Roy (2016, Lemma 3). We then continue to lower bound*

$$(k) = 2\sum_{\tilde{\theta}\in\mathcal{O}_\epsilon} \sum_{\tilde{\theta}^\star\in\mathcal{O}_\epsilon} \mathbb{P}[\tilde{\Theta}=\tilde{\theta}]\mathbb{P}[\tilde{\Theta}^\star=\tilde{\theta}^\star]\big(\mathbb{E}[\phi_\beta(\langle\pi_\star(\tilde{\theta}), \Theta\rangle)|\tilde{\Theta}^\star=\tilde{\theta}^\star] - \mathbb{E}[\phi_\beta(\langle\pi_\star(\tilde{\theta}), \Theta\rangle)]\big)^2$$
$$\geqslant 2L_1^2\sum_{\tilde{\theta}\in\mathcal{O}_\epsilon} \sum_{\tilde{\theta}^\star\in\mathcal{O}_\epsilon} \mathbb{P}[\tilde{\Theta}=\tilde{\theta}]\mathbb{P}[\tilde{\Theta}^\star=\tilde{\theta}^\star]\big(\mathbb{E}[\langle\pi_\star(\tilde{\theta}), \Theta\rangle|\tilde{\Theta}^\star=\tilde{\theta}^\star] - \mathbb{E}[\langle\pi_\star(\tilde{\theta}), \Theta\rangle]\big)^2,$$

*using the fact that $\frac{\phi_\beta(x_1) - \phi_\beta(x_2)}{x_1 - x_2} \geqslant \inf_{x\in[-1,1]} |\phi_\beta(x)'|$ for all $x_1, x_2 \in [-1, 1]$.*

*Combining the inequality on the numerator and the denominator, we get that*

$$\tilde{\Gamma}(\tilde{\Theta}_t^\star, \tilde{\Theta}_t) \leqslant \frac{L_2^2}{2L_1^2} \frac{\left(\sum_{\theta' \in \mathcal{O}_\epsilon} \mathbb{P}[\tilde{\Theta}=\theta']\left(\mathbb{E}[\langle \pi_\star(\theta'), \Theta \rangle | \tilde{\Theta}^\star=\theta'] - \mathbb{E}[\langle \pi_\star(\theta'), \Theta \rangle]\right)\right)^2}{\sum_{\tilde{\theta} \in \mathcal{O}_\epsilon} \sum_{\tilde{\theta}^\star \in \mathcal{O}_\epsilon} \mathbb{P}[\tilde{\Theta}=\tilde{\theta}]\mathbb{P}[\tilde{\Theta}^\star=\tilde{\theta}^\star]\left(\mathbb{E}[\langle \pi_\star(\tilde{\theta}), \Theta \rangle | \tilde{\Theta}^\star=\tilde{\theta}^\star] - \mathbb{E}[\langle \pi_\star(\tilde{\theta}), \Theta \rangle]\right)^2}. \tag{6}$$

*Without loss of generality, we write $\mathcal{O}_\epsilon = \{\tilde{\theta}_1, \ldots, \tilde{\theta}_{|\mathcal{O}_\epsilon|}\}$. Now, we define the random matrix $M \in \mathbb{R}^{|\mathcal{O}_\epsilon| \times |\mathcal{O}_\epsilon|}$ where for each $i, j \in \{1, \ldots, |\mathcal{O}_\epsilon|\}$ the corresponding entry is given by*

$$M_{i,j} = \sqrt{\mathbb{P}[\tilde{\Theta}^\star=\tilde{\theta}_i]\mathbb{P}[\tilde{\Theta}^\star=\tilde{\theta}_j]}\left(\mathbb{E}[\langle \pi_\star(\tilde{\theta}_i), \Theta \rangle | \tilde{\Theta}^\star=\tilde{\theta}_j] - \mathbb{E}[\langle \pi_\star(\tilde{\theta}_i), \Theta \rangle]\right).$$

*We note that we can rewrite eq. (6) using the trace and the Frobenius norm of the matrix $M$, as*

$$\tilde{\Gamma}(\tilde{\Theta}_t^\star, \tilde{\Theta}_t) \leqslant \frac{L_2^2}{2L_1^2} \frac{\text{Trace}(M)^2}{||M||_F} \leqslant \frac{L_2^2}{2L_1^2}\text{Rank}(M),$$

*where the last inequality is obtained from Russo & Van Roy (2016, Fact 10).*

*The proof concludes showing the rank of the matrix $M$ is upper bounded by $d$. For the sake of brevity, we define $\bar{\Theta} := \mathbb{E}[\Theta]$ and $Q_i := \mathbb{E}[\Theta | \tilde{\Theta}^\star=\tilde{\theta}_i]$ for all $i \in \{1, \ldots, |\mathcal{O}_\epsilon|\}$. We then have $\mathbb{E}[\langle \pi_\star(\tilde{\theta}_i), \Theta \rangle | \tilde{\Theta}^\star=\tilde{\theta}_j] = \langle \pi_\star(\tilde{\theta}_i), Q_j \rangle$ and $\mathbb{E}[\langle \pi_\star(\tilde{\theta}_i), \Theta \rangle] = \langle \pi_\star(\tilde{\theta}_i), \bar{\Theta} \rangle$. Since the inner product is linear, we can rewrite each entry $M_{i,j}$ of the matrix $M$ as*

$$\sqrt{\mathbb{P}[\tilde{\Theta}^\star=\tilde{\theta}_i]\mathbb{P}[\tilde{\Theta}^\star=\tilde{\theta}_j]}\langle \pi_\star(\tilde{\theta}_i), Q_j - \bar{\Theta} \rangle.$$

*Equivalently, the matrix $M$ can be written as*

$$\begin{bmatrix} \sqrt{\mathbb{P}[\tilde{\Theta}^\star=\tilde{\theta}_1]}\pi_\star(\tilde{\theta}_i) \\ \vdots \\ \sqrt{\mathbb{P}[\tilde{\Theta}^\star=\tilde{\theta}_{|\mathcal{O}_\epsilon|}]}\pi_\star(\tilde{\theta}_i) \end{bmatrix} \begin{bmatrix} \sqrt{\mathbb{P}[\tilde{\Theta}^\star=\tilde{\theta}_1]}(Q_1 - \bar{\Theta}) & \ldots & \sqrt{\mathbb{P}[\tilde{\Theta}^\star=\tilde{\theta}_{|\mathcal{O}_\epsilon|}]}(Q_{|\mathcal{O}_\epsilon|} - \bar{\Theta}) \end{bmatrix}.$$

*This rewriting highlights that $M$ can be written as the product of a $|\mathcal{O}_\epsilon|$ by $d$ matrix and a $d$ by $|\mathcal{O}_\epsilon|$ matrix and therefore has a rank lower or equal than $\min(d, |\mathcal{O}_\epsilon|)$.*

# F    Regarding the gaps in previous literature

In Section 1, we mentioned that the main results of Dong et al. (2019) are incorrect due to two issues. First, their regret analysis combines results whose hypotheses are not compatible. Second, their analysis of the Thompson Sampling information ratio contains an unaddressed gap for $\beta > 2$. We elaborate on both points below.

## F.1    Regarding the incompatible results

**Short explanation.**    The regret bounds in Dong et al. (2019, Theorems 1 and 5) combine two results with incompatible hypotheses. Specifically, the paper uses a uniform bound on the information ratio of Thompson Sampling (reported in Dong et al. (2019, Appendix B, Eq. (18))) together with Dong & Van Roy (2018, Theorem 1), which requires a uniform bound on the information ratio of *one-step compressed Thompson Sampling*. Since a bound on the Thompson Sampling information ratio does not, in general, imply a bound on the one-step compressed Thompson Sampling information ratio, the hypotheses needed to invoke Dong & Van Roy (2018, Theorem 1) are not satisfied by the argument in Dong et al. (2019). This mismatch is effectively introduced in Dong et al. (2019, Proposition 9), which incorrectly restates Dong & Van Roy (2018, Theorem 4), replacing the assumption required by Dong & Van Roy (2018, Theorem 4) (a uniform bound on the one-step compressed information ratio) by an assumption on the ordinary Thompson Sampling information ratio.

**Detailed explanation.** In their analysis of Thompson Sampling for bandit problems, Russo & Van Roy (2016) introduced the *information ratio*, defined as the ratio between the squared conditional expected instantaneous regret and a measure of information gained from the observed reward. In the original work, as well as in Dong et al. (2019), the information ratio of Thompson Sampling is defined as the random variable

$$\Gamma_t := \frac{\mathbb{E}_t[R(A^\star, \Theta) - R(A_t, \Theta)]^2}{\mathrm{I}_t(A^\star; R(A_t, \Theta), A_t)}, \tag{7}$$

where $A_t = \pi_\star(\Theta_t)$ is the action played by Thompson Sampling and $R(A_t, \Theta)$ is the associated random reward.

Russo and Van Roy use this concept to derive general regret bounds of the form $\sqrt{\Gamma \, T \, \mathrm{H}(A^\star)}$ under an assumption that the information ratio is uniformly controlled (e.g., $\mathbb{E}[\Gamma_t] \leq \Gamma$ for all $t$). This result is restated in Dong & Van Roy (2018, Proposition 1) using a variant of the information ratio defined in terms of $\Theta$ rather than $A^\star$.

**Dong & Van Roy (2018, Proposition 1)** *For any time horizon $T$, the regret of Thompson Sampling is bounded as*

$$\mathbb{E}[\mathrm{Regret}(T)] \leq \sqrt{\Gamma \cdot T \cdot \mathrm{H}(\Theta)},$$

*where $\Gamma$ is a uniform upper bound on the information ratio* $\Gamma_t(\Theta, \Theta_t) := \frac{\mathbb{E}_t[R(A^\star, \Theta) - R(A_t, \Theta)]^2}{\mathrm{I}_t(\Theta; (R(A_t, \Theta), \Theta_t))}.$

A limitation of this approach is that $\mathrm{H}(A^\star)$ (and similarly $\mathrm{H}(\Theta)$) can grow arbitrarily large with the number of actions, or be infinite when the action space is infinite or continuous. The work of Dong & Van Roy (2018) focusses on addressing this limitation. At a high level, their idea is to relate the regret of Thompson Sampling to the regret of an approximation they refer to as *one-step compressed* Thompson Sampling. Under continuity assumptions on the expected reward with respect to the action space, they derive bounds in terms of a *compressed statistic* $\Theta_\varepsilon$ of the parameter $\Theta$, together with the *information ratio of one-step compressed Thompson Sampling*, defined as

$$\tilde{\Gamma}_t(\tilde{\Theta}_t^\star, \tilde{\Theta}_t) := \frac{\mathbb{E}_t\big[R(\pi_\star(\tilde{\Theta}_t^\star), \Theta) - R(\pi_\star(\tilde{\Theta}_t), \Theta)\big]^2}{\mathrm{I}_t\big(\tilde{\Theta}_t^\star; R(\pi_\star(\tilde{\Theta}_t), \Theta), \tilde{\Theta}_t\big)}. \tag{8}$$

Here $\tilde{\Theta}_t^\star$ and $\tilde{\Theta}_t$ are auxiliary random variables constructed from $\Theta_\varepsilon$ and from a statistic $\Theta_{\varepsilon,t}$ derived from the Thompson Sampling sampled parameters $\Theta_t$; see Dong & Van Roy (2018, Proposition 2). The resulting bound, presented in Dong & Van Roy (2018, Theorem 1), is of the form $\varepsilon T + \sqrt{\tilde{\Gamma} \, T \, \mathrm{H}(\Theta_\varepsilon)}$, where $\varepsilon$ controls the quantization level and $\tilde{\Gamma}$ is a uniform upper bound on the *one-step compressed* information ratio $\tilde{\Gamma}_t(\tilde{\Theta}_t^\star, \tilde{\Theta}_t)$.

**Dong & Van Roy (2018, Theorem 1)** *For any time horizon $T$, the regret of Thompson Sampling is bounded as*

$$\mathbb{E}[\mathrm{Regret}(T)] \leq \varepsilon \cdot T + \sqrt{\tilde{\Gamma} \cdot T \cdot \mathrm{H}(\Theta_\varepsilon)},$$

*where $\tilde{\Gamma}$ is a uniform upper bound on $\tilde{\Gamma}_t(\tilde{\Theta}_t^\star, \tilde{\Theta}_t)$ and $\tilde{\Theta}_t^\star$ and $\tilde{\Theta}_t$ are defined as in Dong & Van Roy (2018, Proposition 2).*

It is worth emphasizing that the proof of Dong & Van Roy (2018, Theorem 1), via Dong & Van Roy (2018, Proposition 2), establishes the existence of $\tilde{\Theta}_t^\star$ and $\tilde{\Theta}_t$ but does not provide an explicit construction. Consequently, bounding $\tilde{\Gamma}_t(\tilde{\Theta}_t^\star, \tilde{\Theta}_t)$ outside of specific settings (such as linear bandits) can be technically challenging. In the $d$-dimensional logistic bandit setting, Dong and Van Roy conjecture that $\tilde{\Gamma}_t(\tilde{\Theta}_t^\star, \tilde{\Theta}_t)$ is uniformly bounded by $d/2$; see Dong & Van Roy (2018, Conjecture 1).

**Dong & Van Roy (2018, Conjecture 1)** *Under the logistic bandit setting with logistic function $\phi_\beta(x)$, let $\tilde{\Theta}_t^\star$ and $\tilde{\Theta}_t$ be defined as in Dong & Van Roy (2018, Proposition 2). Then, for any $\beta > 0$,*

$$\tilde{\Gamma}_t(\tilde{\Theta}_t^\star, \tilde{\Theta}_t) \leq \frac{d}{2}.$$

Combining this conjecture with Dong & Van Roy (2018, Theorem 1) yields Dong & Van Roy (2018, Theorem 4).

**Dong & Van Roy (2018, Theorem 4)** *For any $\beta > 0$, and for all $\mathcal{A}, \mathcal{O} \subseteq \mathbf{B}_d(0, 1)$ with minimax alignment constant $\alpha > 0$, under the logistic bandit setting with logistic function $\phi_\beta(x)$, assuming that the one-step compressed information ratio $\tilde{\Gamma}_t(\tilde{\Theta}_t^\star, \tilde{\Theta}_t)$ is uniformly bounded by $\tilde{\Gamma}$ (with $\tilde{\Gamma} = d/2$ according to Dong & Van Roy (2018, Conjecture 1)), the Thompson Sampling regret is bounded as*

$$\mathbb{E}[\text{Regret}(T)] \leqslant 2\sqrt{2\tilde{\Gamma}T \log\left(3 + \frac{6\sqrt{2T}}{d} \cdot \frac{\beta e^{\alpha\beta}}{(1 + e^\beta)^2}\right)}.$$

To apply Dong & Van Roy (2018, Theorem 4), one must establish a uniform bound on the *one-step compressed* information ratio $\tilde{\Gamma}_t(\tilde{\Theta}_t^\star, \tilde{\Theta}_t)$. However, Dong et al. (2019, Proposition 9) instead assumes a uniform bound on the ordinary Thompson Sampling information ratio $\Gamma_t$.

**Dong et al. (2019, Proposition 9)** *For any $\beta > 0$, and for all $\mathcal{A}, \mathcal{O} \subseteq \mathbf{B}_d(0, 1)$ with minimax alignment constant $\alpha > 0$, under the logistic bandit setting with logistic function $\phi_\beta(x)$, assuming that the information ratio of Thompson Sampling $\Gamma_t$ is uniformly bounded by $\bar{\Gamma}$, then the Thompson Sampling regret is bounded as*

$$\mathbb{E}[\text{Regret}(T)] \leqslant 2\sqrt{2\bar{\Gamma}T \log\left(3 + \frac{6\sqrt{2T}}{d} \cdot \frac{\beta e^{\alpha\beta}}{(1 + e^\beta)^2}\right)}.$$

As written, Dong et al. (2019, Proposition 9) is not a valid restatement of Dong & Van Roy (2018, Theorem 4): the latter requires a uniform bound on $\tilde{\Gamma}_t(\tilde{\Theta}_t^\star, \tilde{\Theta}_t)$, not on $\Gamma_t$. Since $\Gamma_t$ and $\tilde{\Gamma}_t$ are distinct quantities, a uniform bound on $\Gamma_t$ does not, in general, imply a uniform bound on $\tilde{\Gamma}_t$ without additional assumptions. This discrepancy undermines the derivation of Dong et al. (2019, Theorem 5) and its corollary Dong et al. (2019, Theorem 1).

Finally, note that one can apply Dong & Van Roy (2018, Proposition 1) directly if one has a uniform control of the ordinary Thompson Sampling information ratio. However, this route yields bounds that scale with $\mathrm{H}(\Theta)$ (or $\mathrm{H}(A^\star)$), which can be prohibitively large or infinite for large, continuous, or otherwise unstructured spaces. For instance, combining Dong & Van Roy (2018, Proposition 1) with our bound on the Thompson Sampling information ratio in Proposition 4 results in a regret bound of order $O(\sqrt{dT \log(|\mathcal{O}|)})$, which becomes vacuous for infinite or continuous parameter spaces.

### F.2 Regarding a gap in the analysis of the information ratio in Dong et al. (2019)

We also identified an issue in the proof of Dong et al. (2019, Theorem 5) (end of the proof on page 20), where the inequality $\chi > \xi > 0.1\lambda$ is stated without justification. This inequality is used to conclude a bound on the Thompson Sampling information ratio, but the only evidence provided is Dong et al. (2019, Figure 3), which plots $\chi(\lambda, \beta)$ and $\xi(\lambda, \beta)$ for the specific case $\beta = 2$. While the plot is suggestive for $\beta = 2$, it does not establish the inequality for the range required by the proof (in particular $\beta \geqslant 2$), nor does it provide a general argument. Moreover, the computation of $\chi(\lambda, \beta)$ and $\xi(\lambda, \beta)$ is nontrivial; we attempted to reproduce Dong et al. (2019, Figure 3) computationally and were unable to do so. Consequently, in the absence of a proof of $\chi > \xi > 0.1\lambda$ for the relevant regime of $\beta$, the argument leading to Dong et al. (2019, (18)), and in particular the step used to control the information ratio for large $\beta$, is not justified as written.

# G    Details on the Numerical Experiments

This appendix provides additional details on the numerical experiments presented in Section 6.

To recall, we consider a synthetic logistic bandit with dimension $d$, horizon $T$, and logistic slope parameter $\beta > 0$. We set the $d$-dimensional unit sphere $\mathbf{S}_d(0,1)$ as both the action space $\mathcal{A}$ and parameter space $\mathcal{O}$, together with a uniform prior on the parameter $\Theta$. We approximate the expected regret using a Monte Carlo method and performed the posterior sampling of the parameter using the Metropolis–Hastings algorithm.

We provide below details regarding:

- the approximation of the expected regret using a Monte Carlo method;

- the sampling of the posterior distribution using the Metropolis–Hastings algorithm;

- the parameters we used for the simulations;

- the accuracy of our approximation.

## G.1    Monte Carlo approximation of axpected regret

To approximate the expected regret of Thompson Sampling, we perform $K$ independent experiments of horizon $T$. At the start of each experiment $k = 1, \ldots, K$ we draw $\theta_k^\star \sim \mathbb{P}(\Theta)$. Then, for each time step $t = 1, \ldots, T$, we sample $N$ parameters $\{\theta_{i,k,t}\}_{i=1}^N \sim \mathbb{P}(\Theta|H^t = h_k^t)$ from the posterior, where $h_k^t$ denotes the history up to time $t$. These samples are used to approximate the conditional instantaneous regret:

$$\mathbb{E}\Big[R(\pi_\star(\Theta), \Theta) - R(\pi_\star(\hat{\Theta}_t), \Theta) \,\big|\, H^t = h_k^t\Big] \;\approx\; \frac{1}{N^2} \sum_{i=1}^N \sum_{j=1}^N \bar{R}(\pi_\star(\theta_{i,k,t}), \theta_{i,k,t}) - \bar{R}(\pi_\star(\theta_{j,k,t}), \theta_{i,k,t}),$$

where $\bar{R}(a, \theta)$ is the *expected reward* for action $a$ and parameter $\theta$, that is $\bar{R}(a, \theta) = \phi_\beta(\langle a, \theta \rangle)$. We then arbitrarly set $\hat{\theta}_{k,t} = \theta_{1,k,t}$ and sample a Bernoulli reward $r_{k,t} \sim \text{Bern}(\phi_\beta(\langle \hat{\theta}_{k,t}, \theta_k^\star \rangle))$ which we use to construct $h_k^{t+1} = h_k^t \cup \{r_{k,t}, \hat{\theta}_{k,t}\}$.

Averaging across $K$ experiments yields an unbiased Monte Carlo estimate of the expected regret at time $t = 1, \ldots, T$, effectively taking the expectation over the history of observations. The approximation of the cumulative expected regret up to time $T$ is then obtained by taking the sum over the time steps $t = 1, \ldots, T$:

$$\mathbb{E}[\text{Regret(T)}] \approx \frac{1}{KN^2} \sum_{t=1}^T \sum_{k=1}^K \sum_{i=1}^N \sum_{j=1}^N \bar{R}(\pi_\star(\theta_{i,k,t}), \theta_{i,k,t}) - \bar{R}(\pi_\star(\theta_{j,k,t}), \theta_{i,k,t}).$$

## G.2    Posterior Sampling via Metropolis–Hastings

Since the logistic likelihood does not admit a closed-form posterior, we use a Metropolis–Hastings sampler with von Mises–Fisher random-walk proposals on the unit sphere. The proposal distribution is centered at the current state with concentration parameter $\kappa$, which we tune to obtain an acceptance rate of about 40%. The acceptance probability reduces to the Metropolis ratio

$$\alpha(\theta \to \theta') = \min\left(1, \; \frac{L(\theta'; h_k^t)}{L(\theta; h_k^t)}\right),$$

where $L(\theta; h_k^t)$ is the logistic likelihood given the history $h_k^t$. At each round, we run $n$ iterations to generate posterior samples.

## G.3    Parameter choices

The parameters used in the numerical experiments are summarized in Table 2. The experiments were run on an NVIDIA RTX 4000 Ada GPU and completed in about 30 minutes.

Table 2: Parameters used in the numerical experiments.

| | |
|---|---|
| Time horizon | $T = 200$ |
| Dimension | $d = 10$ |
| Number of experiments | $K = 1024$ |
| Posterior samples per step | $N = 64$ |
| Initial concentration parameter | $\kappa = 8.0$ |
| MH iterations per step | $n = 16$ |
| PyTorch random seed | $0$ |

## G.4 Accuracy of the simulations and confidence intervals

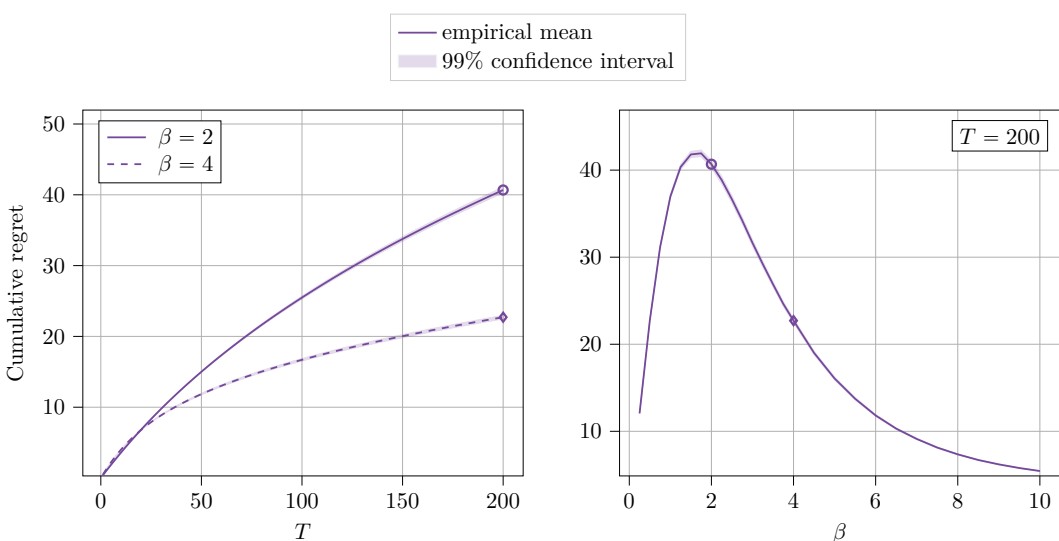

Figure 6: The left sub-figures shows the evolution of the empirical mean and 99% confidence intervals of the expected regret with the time steps $T$ for $\beta \in \{2, 4\}$. The right sub-figure illustrates the behavior of the empirical mean and 99% confidence intervals of the expected regret at time $T = 200$ for values of $\beta$ ranging in $[0.25, 10]$. The confidence interval deviates at most by 1.09% from the empirical mean on the left sub-figure and by 1.13% on the right sub-figure.

