# OpenReview forum: "An Information-Theoretic Analysis of Thompson Sampling for Logistic Bandit Problems"
_TMLR — Accepted by TMLR_

### Review · Reviewer_LMvL · 2025-09-24

**Summary Of Contributions:**

The paper presents an information-theoretic analysis for logistic bandits and derives new regret bounds for Thompson Sampling. The focus is on theory. The main results look solid.

Strengths.

1. Clear problem setup and tight theoretical development.

2. New regret bounds for Thompson Sampling in the logistic setting.

3. Proofs appear careful and technically sound, and the claims are positioned against prior work.

Weaknesses

1. Numerical experiments are not discussed in depth. How many replications did you choose? Can you also show the confidence intervals?

**Audience:**

Yes

**Audience Explanation:**

Those working on bandit algorithms, reinforcement learning theory, and information-theoretic methods, would likely be interested in these findings, since they provide new regret bounds for TS in the logistic bandit setting.

**Broader Impact Concerns:**

I did not find any broader impact concerns.

**Claims And Evidence:**

Yes

**Claims Explanation:**

Yes. The theoretical claims are supported by rigorous proofs and are presented in a clear manner, so the main evidence is convincing from a theory standpoint. While the paper does not provide extensive numerical experiments, the mathematical analysis itself is solid.

**Requested Changes:**

Numerical experiments

1. How many replications did you choose? Can you also show the confidence intervals? This will strengthen your work.
2. Fig. 3: why is the regret for beta>4 smaller than for beta=2 under TS? A short explanation would help.
3. Can you (anonymously for now) share your code to justify reproducibility?

Introduction

4. Table 1: I compared your Table 1 with Faury et al. (2020), *Improved Optimistic Algorithms for Logistic Bandits*. Can you explain the relationship between their kappa and your e^(beta)? The rate of GLM-TSL, and Logistic-UCB-2 appears in both their paper and your work, but it looks not very consistent to me.

Problem Setup

5. Could you clarify what asymptotics your regret bound refers to? In particular, which parameter is taken to infinity in your analysis? For example, is the bound derived in the standard bandit sense where the time horizon T, or does it involve the dimension d,  the parameter norm beta, or some other scaling? A short statement of what limit your results assume would make the presentation clearer.

6. In your setup, beta is a scale parameter known to the agent. Does this mean beta is fixed in the analysis, and if so, how exactly does it help in the regret bound (e.g., avoiding exponential terms like e^(beta))?

---

> ### Author Response · Authors · 2025-09-30
> **Authors response**
>
> We thank the reviewer for their careful reading and insightful feedback. We are grateful that the review highlights our results as new and solid, our proofs as rigorous and convincing, and our findings as of interest to the RL community. We address the comments below.
>
> ---
>
> ### A. Numerical experiments
> ---
>
> **1. Regarding the number of replications and the confidence intervals**
>
> We averaged over $2^{18} \approx 4.2$ million replications to estimate the expected regret of Thompson Sampling. Following the reviewer’s suggestion, we now include figures with 99% confidence intervals in the revised manuscript (see the new Appendix G.). These intervals are extremely tight (never exceeding ±1.13% of the mean), and thus nearly indistinguishable from the plotted empirical regret curves. Appendix G also provides additional details on the Monte Carlo approximation, the Metropolis–Hastings posterior sampling procedure, and the full set of experimental parameters.
>
> **2. Remark about the expected regret decreasing with $\beta > 2$**
>
> In Figure 3, we observe that the expected regret decreases once $\beta$ becomes sufficiently large (e.g., $\beta > 2$). This is consistent with the intuition discussed in the Introduction (page 1): as $\beta$ increases, the logistic function becomes steeper, which sharpens the distinction between near-optimal and suboptimal actions. In this regime, Thompson Sampling can more easily separate good actions from bad ones, which accelerates learning and leads to lower cumulative regret. We have updated Section 6 (*Numerical Experiment*) to include a remark on this observation.
>
> **3. Reproducibility and code release**
>
> We have uploaded our code as a `.zip` archive in the *Supplementary Material*. It contains:
> - a `README.md`;
> - `logistic_bandits_ts.py` (Thompson Sampling);
> - `mh_sphere.py` (Metropolis–Hastings on the sphere);
> - `plots_ts.py` (plotting utilities);
> - and pre-computed results and figures.
>
> After the review process, we will release the repository on GitHub and link it in the paper.
>
> ---
>
> ### B. Introduction
> ---
>
> **4. On the relationship between $\kappa$ in Faury et al. (2020) and our $e^{\beta}$**
>
> The two notations arise from different but equivalent definitions of the logistic model.
>
> In our paper, we use a slope parameter $\beta > 0$:
>
> $\phi_\beta(\langle a , \theta \rangle) = \frac{1}{1 + \exp(-\beta <a, \theta >)}, \quad a,\theta \in \mathbf{B}_d(0,1).$
>
> By contrast, Faury et al. (2020) employ the unit-slope logistic link but allow $\theta$ to vary in a ball of radius $S$:
>
> $\mu(\langle a , \theta \rangle) = \frac{1}{1 + \exp(-<a, \theta >)}, \quad a \in \mathbf{B}_d(0,1),\ \theta \in \mathbf{B}_d(0,S).$
>
> These are equivalent under $S = \beta$; assuming $\theta \in \mathbf{B}_d(0,1)$ with slope $\beta$ is the same as $\theta \in \mathbf{B}_d(0,S)$ with unit slope, where $S=\beta$. This is a matter of notation only and does not affect the asymptotic form of the regret bounds.
>
> Regarding $\kappa$, Faury et al. define
>
> $\kappa = \inf_{a \in \mathbf{B}_d(0,1),\ \theta \in \mathbf{B}_d(0,S)} \frac{1}{\mu'(< a,\theta>)}.$
>
> They show (in Section 2) that $\kappa = O(e^S)$. Consequently, their regret bound for Logistic-UCB-2 can be written as
>
> $O\left(d T^{1/2}\log T + \underbrace{e^S}_{\kappa\ \text{term}} d^2 \log^2 T \right),$
>
> which in our parameterization corresponds to
>
> $O\left(d T^{1/2}\log T + e^\beta d^2 \log^2 T \right).$
>
> ---
>
> ### C. Problem Setup
> ---
>
> **5. On the scaling interpretation of our regret bound**
>
> Our regret inequality is fully nonasymptotic: it holds for all $T \geq 1$, $d \geq 1$, $\beta > 0$, and $\alpha \in (0,1]$. We do not assume any parameter tends to infinity. The big-$O$ notation was only to highlight scaling with $T$, $d$, $\alpha$, and $\beta$.
>
> **6. On the role of the parameter $\beta$ in our analysis**
>
> In our setup, $\beta > 0$ is a fixed scale parameter of the logistic model and assumed known to the agent. This is standard in the logistic bandit literature.
>
> The improvement in our regret bound comes from new proof techniques for bounding the information ratio of Thompson Sampling:
> - Relating regret and information gain to the *expected variance of the regret* (conditioned on the sampled parameter), rather than the expected regret itself (as in Dong & Van Roy, 2018).
> - Observing that the limit case $\beta \to \infty$ can serve as a uniform upper bound, simplifying the analysis.
>
> Together, these steps avoid scaling with the inverse of the smallest slope of the logistic function (which grows exponentially in $\beta$), ensuring that the final regret bound depends only on $d$ and the minimax constant $\alpha$, with no dependence on $\beta$.
>
> ---
>
> We thank the reviewer once again for their thoughtful comments, which helped us improve both the clarity of the paper and the presentation of our results. We hope our responses address the concerns and we are happy to provide further clarifications if needed.

---

> > ### Comment · Reviewer_LMvL · 2026-01-24
> > **Thanks the authors for the response**
> >
> > Thanks for the response. Your response is visible to me only after today, but it indeed makes much more sense to me.

---

### Review · Reviewer_aXzJ · 2025-10-05

**Summary Of Contributions:**

In this paper, the authors present a novel analysis for the Thompson Sampling algorithm (TS) for logistic bandit problems, in which one must sequentially select actions from an action space and receive rewards with probability proportionate to the chosen action's alignment to an unknown parameter vector. Their new information theoretic regret bound significantly improves upon previous ones by improving the scaling with respect to the "scale parameter", $\beta$ of the reward distribution from exponential to logarithmic, and removing the dependence on the cardinality of the action set. Additionally, their regret bound does not depend on the "fragility dimension", a measure that frequents previous results, which can grow exponentially with the problem dimension. To their knowledge, this is the first such regret bound for any algorithms for logistic bandit problems.

The core of the authors' results rely on the "information ratio", a measure defined as the ratio between the squared expected regret of the algorithm's choice, and the information about the unknown environment parameter gained by it, quantifying the tradeoff between exploration and exploitation. By upper-bounding the information ratio of TS, one can argue that an action that incurs regret is compensated by large information gain. To obtain such an upper bound, the authors first bound both the regret and the information gain in terms of the variance of the regret, as opposed to the expectation as in prior work, avoiding the exponential scaling with $\beta$, and then show that it suffices to prove the special case of $\beta \to \infty$, thus simplifying the analysis.

**Audience:**

Yes

**Audience Explanation:**

The paper presents novel and improved bounds on the regret of Thompson Sampling on logistic bandit problems - both of which very well-celebrated concepts that are quite conceptually simple. Developing such improved bounds, especially one that breaks the dependence on some parameters, whether down from exponential or eliminating completely, would serve to explain the empirical effectiveness of these algorithms and even inspire in their methodology further exploration across the field of machine learning theory on other algorithms and/or problems. From a theorist's perspective, the results presented are those I would like my peers to know about.

It can be argued that the paper would benefit from an overview of the real-world/industrial applications of Thompson Sampling and/or logistic bandits. Regardless, theoretical advancement in the form of upper/lower bounds are important.

**Broader Impact Concerns:**

The work is purely analytical and theoretical and does not require a broader impact statement in my opinion.

**Claims And Evidence:**

Yes

**Claims Explanation:**

To my knowledge, the authors' proofs and claims are supported by sound mathematical evidence. Their results are somewhat surprising - that it is possible to get rid of the dependencies (exponential for $\beta$ and completely for the cardinality of the set of actions), but the authors' comments in section 4 and 5 supplied good evidence and intuition as to why the readers should believe that their claims are correct. In particular, most of the important claims, theorems, lemmas, and definitions are supplied with textual explanations of their role in the grand scheme of the proof, and the intuitions required to follow the flow of logic, which is a quality I think should be appreciated in academic publications.

**Requested Changes:**

As noted above, the paper does not make much statements about the applicational advantage and implications of their improved analytical bounds. While not as necessary as the main results, I would appreciate some more references to industrial applications of Thompson Sampling and/or logistic bandits, as well as discussions about the implications of the authors' results and bounds to application, if any and if applicable.

---

> ### Author Response · Authors · 2025-10-06
> **Authors response**
>
> We would like to thank the reviewer for taking the time to review our paper and for providing us with insightful feedback. We are grateful that the review highlights the novelty, rigor, and clarity of our analysis, as well as the significance of our results for understanding the behavior of Thompson Sampling in logistic bandit problems.
>
> ---
>
> The reviewer suggested expanding the discussion on the applications and implications of our results. We agree that this addition strengthens the paper and have accordingly revised the introduction and the conclusion.
>
> In particular, at the beginning of the introduction, following the existing sentence:
>
> > *This setting is used to model various scenarios, such as click-through rate prediction, spam email detection, and personalized advertisement (Chapelle & Li, 2011; Russo & Van Roy, 2018).*
>
> we added:
>
> > *Beyond these examples, bandit-based decision systems are also used in news and content recommendation, dynamic pricing, information retrieval, and healthcare (see Bouneffouf et al., 2020 for a survey of the different applications, and Li et al., 2010 for a large-scale deployment).*
>
> We also added the following sentence at the end of the introduction:
>
> > *These results help to explain the empirical performance of Thompson Sampling in logistic bandit problems across different logistic regimes and for large or continuous action spaces, and they highlight the importance of alignment between the action and parameter spaces.*
>
> Finally, we included a similar sentence in the conclusion to reinforce this connection:
>
> > *Overall, these results help to explain the empirical performance of Thompson Sampling in logistic bandit problems across different logistic regimes and for large or continuous action spaces, and highlight the importance of alignment between the action and parameter spaces.*
>
> These modifications can be found in the revised PDF and are written in dark green for ease of review (as are the changes made in response to earlier feedback).
>
> ---
>
> We thank the reviewer once again for their thoughtful comments and positive evaluation of our work.

---

### Review · Reviewer_WQ2H · 2025-10-31

**Summary Of Contributions:**

This paper studies the performance of the Thompson Sampling(TS) algorithm in logistic bandit problems using an information-theoretic framework. The main theoretical result shows that the expected regret of TS is bounded by $O(\frac{d}{\alpha}\sqrt{T\log(\beta T/d)})$, where $d$ is the problem dimension and $\alpha$ is a minimax alignment constant characterizing the geometry between the action and parameter spaces. This paper also includes numerical experiments validating the improved regret bounds.

Strengths:

1.	Result improvement: It provides a regret bound for logistic bandits that avoids exponential dependence on the logistic slope parameter $\beta$ and does not depend on the number of actions.
2.	Technical innovation: It introduces a refined analysis of the information ratio that lower bounds information via the variance of regret rather than its expectation.

Weaknesses:

1.	The paper introduces the minimax alignment constant $\alpha$ as a key geometric quantity connecting the action and parameter spaces (Sec.2).  Given its central role in the regret upper bound, the authors may provide a more detailed discussion of its practical interpretation and implications.
2.	Section 5 is mathematically dense and relies on multiple non-trivial lemmas, which may make the logical flow difficult to follow for readers who are not deeply familiar with information-theoretical analyses of bandit algorithms.
3.	Is there a lower bound in the considered setting? Can authors discuss the optimality of  the obtained upper bound in terms dependence on $d$, $\alpha$, $\sqrt{T}$ and $\log (\beta T/d)$?

**Additional Comments:**

NA

**Audience:**

Yes

**Audience Explanation:**

TS and bandit algorithms are of broad interest in online learning and reinforcement learning.

**Claims And Evidence:**

Yes

**Claims Explanation:**

The theoretical claims are supported by formal statements and complete proofs (Theorem 2; Proposition 4; Theorem 5, with auxiliary lemmas in Appendices A–D), and the empirical section illustrates the scaling predicted by the theory (Section 6, Figure 3).

**Requested Changes:**

Please see the summary part.

1.	Since $\alpha$ plays a central role in regret upper bound, it would be helpful to include more intuition or examples illustrating its practical meaning.
2.	The authors could improve accessibility by offering more intuitive explanations of key steps of highlighting the main insights behind the lemmas, helping readers better grasp the logic.

---

> ### Author Response · Authors · 2025-11-07
> **Authors response**
>
> We thank the reviewer for their time and constructive feedback. We appreciate that they found our results to be of broad interest, supported by formal statements and complete proofs, as well as contributing improved regret bounds and a refined analysis of the information ratio. We address the reviewer’s specific comments below.
>
> ----
> **Comment 1:** *"Since $\alpha$ plays a central role in regret upper bound, it would be helpful to include more intuition or examples illustrating its practical meaning."*
>
> We thank the reviewer for the suggestion and have added additional explanations to help readers build intuition about $\alpha$ and understand why it appears in the analysis. The added paragraphs are provided below.
>
> >**[Sec 2]** [...] Intuitively, $\alpha$ quantifies *how well* the actions set covers for any of the possible parameters: a value of $\alpha$ close to $1$ indicates that, for any parameter, there exists an action that aligns well with it, whereas a value close to $0$ means that there exists a parameter that is at best nearly orthogonal to all action. [...]
>
> >**[Sec 4, after Proposition 4]** The dependence on $1/\alpha$ should not be surprising: when the action and parameter spaces are poorly aligned, that is when $\alpha$ is small, it is possible to construct an action space $\mathcal{A}$ and a parameter space $\mathcal{O}$ such that the parameters are nearly orthogonal to all actions (see Dong et al. (2019, Appendix D)). In such a case, the reward probabilities associated with different actions become almost indistinguishable, so observing the
> reward provides little information about the environment, which in turn leads to larger information ratios.
> ---
> **Comment 2:** *"The authors could improve accessibility by offering more intuitive explanations of key steps of highlighting the main insights behind the lemmas, helping readers better grasp the logic."*
>
> We appreciate the reviewer’s suggestion. We have added further explanations in Sec 5 to clarify our reasoning. We also included additional intuition on how Lemma 14 and 15 are used in the proof. We believe these additions help communicate the core ideas underlying the analysis.
>
> >**[Sec 5.1]** Intuitively, if the expected variance of reward probability is high, the agent is still exploring new actions and gathering information about the parameter $\Theta$. In such cases, although the instantaneous regret may remain large, the agent continues to learn about the environment, which can allow the information ratio to be controlled. If the expected variance of reward probability is low, we need to show that the agent has identified near-optimal actions and is now incurring small instantaneous regrets. Thus the next part of the proof, Section 5.2, will be focused on making this intuition rigourous by relating the expected instantaneous regret to the expected variance of reward probability.
>
> >**[Sec 5.2]** The two lemmata above will be key to relating the expected instantaneous regret to the expected variance of the reward probability. After conditioning on the sampled TS parameter $\hat{\Theta}$, we will express the expected regret as a function of $\hat{\Theta}$ and apply Lemma 15 to relate it to the regret variance. We will then use Lemma 14 to decouple inner-product terms and reveal the dependence on the dimension $d$. Before doing so, however, we introduce additional notation and a few reformulations to make Lemma 15 applicable.
> ---
> **Comment 3:** *"Is there a lower bound in the considered setting? Can authors discuss the optimality of the obtained upper bound in terms dependence on $d$, $\alpha$, $\sqrt{T}$ and $\log (\beta T/d)$?"*
>
> We thank the reviewer for this question. The work of Abeille et al. (2021) establishes a problem-dependent lower bound for logistic bandits of order $O(d\sqrt{T\kappa^{-1}})$ where $\kappa^{-1}$ scales with the smallest slope of the logistic function. This implies that, up to logarithmic and problem-dependent constants, our bound is tight in its dependence on both $d$ and $T$. We also note that the dependence on $d$ and $T$ matches the minimax lower bound for linear bandits (see Dani et al. (2008)).
>
> Regarding the dependence on $\alpha$, Proposition 11 from Dong et al. 2019 shows that there cannot exist an $\alpha$-independent upper bound that is polynomial in $d$ and sublinear in $T$. Moreover, the scaling in $\alpha^{-1}$ is consistent with the *"worst-case"* construction discussed in Dong et al. (2019, Appendix D). Finally, we note that the logarithmic factor $\log(T/d)$ commonly appears in Bayesian regret upper bounds for linear bandits. Whether this dependence, as well as that on $\log(\beta)$, can be eliminated remains an open question. We have added in Section 4 two paragraphs discussing the dependencies on $d$, $T$, and $\alpha$.
>
> ----
> We thank the reviewer once again for their thoughtful comments. We hope our response addresses all concerns, and are happy to provide further clarifications if needed.

---

### Comment · Action_Editor_fPMR · 2026-01-08
**Please better clarify the claimed incorrectness in Dong et al. (2019)**

Dear authors,

Thanks a lot for submitting this paper to TMLR. In your submission, you claimed that there are incorrect results in Dong et al. (2019), and explained it in both the introduction and Appendix F. Might you clarify your claims with more details? Specifically,

- Please explain the claimed incorrectnesses **in more details, and be technical**. For instance, what are the differences between the uniform bound on information ratio of the "standard" TS and the uniform bound on information ratio of the "one-step compressed" TS? Why Proposition 9 in Dong et al. (2019) is an incorrect restatement of Theorem 4 of Dong & Van Roy (2018)?

- **Please explain why you believe these incorrectnesses cannot be easily fixed.**

Thanks! I am looking forward to learning from you and getting to the bottom of this.

---

> ### Comment · Editors_In_Chief · 2026-01-13
>
> Dear authors,
>
> Please respond to the AE's question soon.
>
> Thank you!

---

> ### Author Response · Authors · 2026-01-16
> **Response to Action Editor**
>
> Dear Action Editor,
>
> Thank you for the careful reading. We have expanded *Appendix F*, adding a *Detailed explanation* paragraph that provides definitions and further technical explanation about the incompatibility issue that arises in Dong et al. (2019).
>
> Below we summarize the key points.
>
> ---
>
> #### 1. Standard TS information ratio vs one-step compressed TS information ratio
>
> In Appendix F, we recall that the  information ratio of Thompson Sampling is defined as
> $\Gamma_t \coloneqq
> \frac{\mathbb{E}_t\left[ R(A^\star,\Theta)-R(A_t,\Theta)\right]^2}
> {I_t\left(A^\star; R(A_t,\Theta),A_t\right)}$ ,
> (or equivalently variants conditioning on $\Theta$, as in Dong & Van Roy (2018, Prop. 1)).
>
> By contrast, Dong & Van Roy (2018) introduce  information ratio of the *one-step compressed Thompson Sampling* as the random variable
> $\tilde{\Gamma}_t(\tilde{\Theta}_t^\star,\tilde{\Theta}_t) \coloneqq \frac{\mathbb{E}_t\left[R(\pi(\tilde{\Theta}_t^\star),\Theta)-R(\pi(\tilde{\Theta}_t),\Theta)\right]^2}{I_t\left(\tilde{\Theta}_t^\star;R(\pi(\tilde{\Theta}_t),\Theta),\tilde{\Theta}_t\right)}$
> .
>
> Here $\tilde{\Theta}^\star_t$ and $\tilde{\Theta}$ are carefully crafted auxiliary random variables whose existence is established in Dong & Van Roy (2018, Prop. 2), constructed respectively from the compressed statistic $\Theta_\varepsilon$ and a equivalent statistic $\Theta_{\varepsilon,t}$ of the TS sampled parameter $\Theta_{t}$. Importantly, these variables are *not* the same as those appearing in *standard* TS information ratio.
>
> As a result, a uniform bound on the standard TS information ratio $\Gamma_t$ does **not** imply a uniform bound on the one-step compressed information ratio $\tilde{\Gamma}_t$.
>
> ---
>
> #### 2. Why Proposition 9 (Dong et al., 2019) is an incorrect restatement of Theorem 4 (Dong & Van Roy, 2018)
>
> Dong & Van Roy (2018, Thm. 4) is obtained by combining their Theorem 1 with their Conjecture 1, and it explicitly assumes a uniform bound on the *one-step compressed* information ratio $\tilde{\Gamma}_t(\tilde{\Theta}_t^\star,\tilde{\Theta}_t)$.
>
> However, Dong et al. (2019, Prop. 9) which restates Dong & Van Roy (2018, Thm. 4) miswrote this assumption and instead required a uniform bound on the *standard* Thompson Sampling information ratio $\Gamma_t$. Since $\Gamma_t$ and $\tilde{\Gamma}_t$ are distinct quantities involving different random variables and different regrets and information terms, this substitution is not justified. Consequently, the hypotheses required to employ Dong & Van Roy (2018, Thm. 4) are not satisfied by the argument in Dong et al. (2019).
>
> This incompatibility is the reason why the regret bounds claimed in Dong et al. (2019, Thms. 1 and 5) are not established as written.
>
> ---
>
> #### 3. Why this issue is not easily fixed
>
> The compression step in Dong & Van Roy (2018) was introduced specifically to avoid entropy terms such as $H(A^\star)$ or $H(\Theta)$, which can be arbitrarily large or infinite in large or continuous action spaces. Their regret bound (Thm. 1) therefore crucially relies on controlling $\tilde{\Gamma}_t$.
>
> However, $\tilde{\Theta}_t^\star$ and $\tilde{\Theta}_t$ are only shown to exist, and no general constructive form is provided. Bounding $\tilde{\Gamma}_t$ beyond special cases is therefore challenging (hence Dong & Van Roy (2018, Conjecture 1)). Replacing $\tilde{\Gamma}_t$ by $\Gamma_t$ is not valid, and reverting to the original regret bound, Russo & Van Roy (2015, Proposition 1) or Dong & Van Roy (2018, Proposition 1)) reintroduces the entropy dependence that compression was designed to eliminate.
>
> ---
>
> #### 4. Additional gap for large $\beta$
>
> In Appendix F we also clarify a second issue in Dong et al. (2019): the inequality $\chi > \xi > 0.1\lambda$ is asserted without proof in the final step of the proof of Theorem 5. This inequality is essential to derive equation (18) and to control the information ratio for large $\beta$. The only evidence provided is a plot for $\beta=2$, which does not establish the inequality for the regime required by the proof. Without this inequality, the derivation of equation (18) for large $\beta$ is not justified as written.
>
> ---
>
> We hope that the expanded Appendix F and the clarifications above address the request for technical detail and make the logical issues transparent.

---

### Decision · Action_Editor_fPMR · 2026-01-08

**Recommendation:** Accept with minor revision

**Additional Comments:**

- Please better clarify the claimed incorrectness in Dong et al. (2019), by replying my comment below.

- Please address the concerns by Reviewer LMvL by replying their comments, specifically:

"Could you clarify what asymptotics your regret bound refers to? In particular, which parameter is taken to infinity in your analysis? For example, is the bound derived in the standard bandit sense where the time horizon T, or does it involve the dimension d, the parameter norm beta, or some other scaling?"

"In your setup, beta is a scale parameter known to the agent. Does this mean beta is fixed in the analysis, and if so, how exactly does it help in the regret bound (e.g., avoiding exponential terms like e^(beta))?"

- When submitting the next version of the paper, please make sure that you have made all the promised changes.

**Audience:**

Yes

**Audience Explanation:**

Theorists in the field of bandit and reinforcement learning will be interested in knowing the findings of this paper.

**Claims And Evidence:**

Yes

**Claims Explanation:**

This paper is mainly theoretical, and its main contribution is to develop novel and better information-ratio bound and regret bound for Thompson sampling (TS) in logistic bandits. Both the analysis techniques and the results are novel and highly non-trivial, and the main result is a significant improvement compared to existing literature. All three reviewers have given Yes to Claims and Evidence, after reading this paper, I agree with them.

However, this paper has claimed a previous paper, Dong et al. (2019), has incorrect results. By reading this paper, I still do not fully understand the claimed incorrectness, and have left a comment asking the authors to further clarify it. **I will NOT recommend the final acceptance until we get to the bottom of it.**

---

> ### Author Response · Authors · 2026-01-24
> **Response to Decision by Action Editor**
>
> Dear Action Editor,
>
> Thank you very much for the positive assessment and for recommending acceptance with minor revision. We believe the requested clarifications, namely, the detailed explanation of the incorrectness in Dong et al. (2019), as well as the responses to Reviewer LMvL regarding the asymptotic interpretation of the regret bound and the role of the parameter $\beta$, have already been addressed in our replies on OpenReview and incorporated into the uploaded revised manuscript (in Appendix F). It is possible that these replies were missed, or that we did not set the appropriate visibility settings (in particular for our response to Reviewer LMvL), if so, we apologize for the confusion.
>
> To ensure everything is clear, we summarize the main points again below for convenience. For completeness, we also invite you to consult the more detailed replies we previously posted in the discussion below, which provide additional context and technical details.
>
> ---
>
> ### Summary of the main points
> ---
>
> #### 1. Standard TS vs one-step compressed TS information ratios.
>
> In Appendix F, we recall that the *standard* TS information ratio is defined as $\Gamma_t \coloneqq \frac{\mathbb{E}_t\left[ R(A^\star,\Theta)-R(A_t,\Theta)\right]^2} {I_t\left(A^\star; R(A_t,\Theta),A_t\right)}$ , (or equivalently variants conditioning on $\Theta$, as in Dong & Van Roy (2018, Prop. 1)). By contrast, Dong & Van Roy (2018) introduce the information ratio of the *one-step compressed Thompson Sampling* as the random variable
>  $\tilde{\Gamma}_t(\tilde{\Theta}_t^\star,\tilde{\Theta}_t) \coloneqq \frac{\mathbb{E}_t\left[ R(\pi(\tilde{\Theta}_t^\star),\Theta)-R(\pi(\tilde{\Theta}_t),\Theta) \right]^2} {I_t\left( \tilde{\Theta}_t^\star; R(\pi(\tilde{\Theta}_t),\Theta),\tilde{\Theta}_t \right)}$ .
>
> Here $\tilde{\Theta}^\star$ and $\tilde{\Theta}$ are carefully crafted auxiliary random variables constructed respectively from the compressed statistic $\Theta_\varepsilon$ and an equivalent statistic $\Theta_{\varepsilon,t}$ of the TS sampled parameter $\Theta_t$. Importantly, these variables are *not* the same as those appearing in the *standard* TS information ratio. As a result, a uniform bound on the standard TS information ratio $\Gamma_t$ does **not** imply a uniform bound on the one-step compressed information ratio $\tilde{\Gamma}_t$.
>
> ---
>
> #### 2. Why Proposition 9 in Dong et al. (2019) is incorrect.
>
> Dong & Van Roy (2018, Thm. 4) explicitly requires a uniform bound on the one-step compressed information ratio $\tilde{\Gamma}_t$. However, Dong et al. (2019, Prop. 9) miswrote this assumption and replaced it with a uniform bound on the standard TS information ratio $\Gamma_t$. Since these are different objects, this substitution is not justified, and the hypotheses required to apply Dong & Van Roy (2018, Thm. 4) are not met. This is why the regret bounds claimed in Dong et al. (2019, Thms. 1 and 5) are not established as written.
>
> ---
>
> #### 3. Why this issue is not easily fixed.
>
> The compression step was introduced precisely to avoid entropy terms such as $H(A^\star)$ or $H(\Theta)$, which can be infinite in large or continuous spaces. The resulting regret bound relies on controlling $\tilde{\Gamma}_t$, but the auxiliary variables defining it are only shown to exist, with no general constructive form. Bounding $\Gamma_t$ alone does not resolve this, while reverting to standard TS bounds reintroduces the problematic entropy dependence.
>
> ---
>
> #### 4. Additional gap for large $\beta$.
>
> Dong et al. (2019) also rely on the unproven inequality $\chi > \xi > 0.1\lambda$ in the final step of the proof of Theorem 5 to derive equation (18) and control the information ratio for large $\beta$. The only evidence provided is a plot for $\beta=2$, which does not establish the inequality in the required regime. Without it, the argument for large $\beta$ is not justified as written.
>
> ---
>
> #### 5. Addressing Reviewer LMvL’s comments
>
> The comment raised by Reviewer LMvL regarding the interpretation of asymptotics and the role of $\beta$ were already addressed in our earlier reply (now visible to everyone).
>
> Briefly, our regret bound is fully nonasymptotic and holds for all $T \ge 1$, $d \ge 1$, $\beta > 0$, and $\alpha \in (0,1]$. The use of big-$O$ notation is purely to summarize scaling. Moreover, while $\beta$ is assumed known (this is standard in logistic bandits), the improvement in our bound does not rely on fixing $\beta$, but rather on new information-ratio techniques that avoid exponential dependence on $\beta$.
>
> ---
>
> All the above clarifications are already reflected in the latest uploaded revision of the paper.
>
> We would like to thank you again for the positive decision and for the careful handling of the manuscript. We hope that the expanded Appendix F and the clarifications above fully address the remaining questions and make the logical issues transparent. We remain happy to answer any further questions if needed.

---

> > ### Comment · Action_Editor_fPMR · 2026-01-27
> > **Thanks!**
> >
> > Dear Authors,
> >
> > Thanks a lot for your detailed and clear explanations! Your points make sense to me. I have also checked the relevant analyses in Dong & Van Roy (2018) and Dong et al. (2019). To the best of my knowledge, I think your arguments above are correct.
> >
> > Thanks!

---

> > > ### Author Response · Authors · 2026-02-01
> > > **Thank you**
> > >
> > > Dear Action Editor,
> > >
> > > Thank you for taking the time to review our explanations and for your kind message.
> > > Please let us know if there is anything further needed from our side.